# Sustainable Dyeing and Functional Finishing of Cotton Fabric by *Rosa canina* Extracts

Raziye Atakan *, Inés Martínez-González, Pablo Díaz-García and Marilés Bonet-Aracil

Department of Textile and Paper Engineering, Universitat Politècnica de València, 03801 Alcoy, Spain; inmargon@epsa.upv.es (I.M.-G.); pdiazga@txp.upv.es (P.D.-G.); maboar@txp.upv.es (M.B.-A.)
* Correspondence: raziyeatakan@gmail.com; Tel.: +34-674435831

**Abstract:** This paper presents a comprehensive study on a novel ultrasound-assisted extraction process for *Rosa canina*, utilizing both dry and fresh fruits, and explores the potential application of *Rosa canina* extraction as a natural dye and functional agent for cotton fabrics. The ultrasound-assisted extraction employed different solvents, including distilled water, methanol, and a water/methanol mixture (50/50% *v/v*), conducted at 60 °C for 60 min. The extracted compounds were characterized through ultraviolet–visible (UV-Vis) spectroscopy and high-performance liquid chromatography with ultraviolet spectroscopy (HPLC-UV) analysis to assess the chemical composition. Textile applications were then performed using bio-mordant chitosan in a pre-mordanting process, and the treated cotton fabrics underwent analysis for surface chemistry and chemical composition using Fourier-transform infrared spectroscopy (FTIR). Untreated and treated fabrics, both with and without mordant, were evaluated for their UV protection and antibacterial properties. Color measurements and dyeability properties of the extractions were also assessed. Furthermore, waste solutions from textile applications were analyzed by UV-Vis spectroscopy to investigate the potential transfer of active compounds to the fabrics. Results indicate that *Rosa canina*, as a plant-based extract, holds significant potential for sustainable dyeing and functional finishing of cotton fabrics.

**Keywords:** natural extracts; dyeing; natural dyes; eco-friendly textiles; sustainable dyeing and functionalization

## 1. Introduction

With growing environmental awareness, there is a heightened interest in the textile industry towards bioactive extracts sourced from plants as renewable and sustainable bio-resources. This interest is driven by the non-toxic, degradable, and eco-friendly nature of these extracts. Much research indicates that natural bioactive extracts offer various functionalities, including antibacterial and antioxidant properties, UV protection, flame retardancy, and more. These findings suggest promising opportunities for the integration of bioactive extracts into textile product development, particularly in the realm of hygiene-related and medical textiles. Furthermore, utilizing natural functional dyes in textiles merges dyeing and finishing procedures, representing an efficient technique characterized by minimal water and energy consumption. Moreover, industrial-scale natural finishing of textiles is emerging as a tangible prospect in the eco-friendly textiles market, paving the way for a greener and more sustainable textile industry [1].

The therapeutic properties of numerous plants have been employed since ancient times. In recent years, natural extracts based on plants and some other natural resources in textile finish have gained significant momentum because of being the best eco-friendly alternatives to synthetic chemicals as dyeing and finishing agents. In addition, most of them are sustainable, renewable, biodegradable, biocompatible, and available in abundance. Some add multifunctionality to textile substrates while contributing to a reduction in carbon emissions and effluent load [2]. Their waste dye solutions also help minimize the discharge of harmful chemicals into water bodies, reducing environmental pollution.

From the perspective of sustainability, textile effluent, which includes many pollutants like reactive dyes, chemicals, high chemical oxygen demand (COD), biological oxygen demand (BOD), and organic compounds, is a major contributor to environmental harm and human health issues [3]. Approximately 40% of colorants used worldwide contain organically bound chlorine, a recognized carcinogen. The organic substances in textile industry wastewater pose significant challenges in water treatment, particularly due to their reactivity with various disinfectants, especially chlorine. These chemicals can either evaporate into the air, leading to respiratory exposure, or be absorbed through the skin, resulting in allergic reactions and potential harm, even affecting children before birth [4].

Compounds such as phenolic and polyphenols [5–9], tannins [10–15], flavonoids [10–12, 16–20], flavones [21], terpenoids [22,23], phenolic acids [18], polypeptides [24], quinines [9,25], flavanols [9,26], coumarins [27], essential oils [18], alkaloids [19,28], lectins [29], carotenoids [28], and polyacetylenes [28] are the most common components extracted from plants. Some of these compounds serve the function of defense against microbes, insects, and herbivores. In contrast, some give flavor, odor, and colored pigments to the plant [30]. Plant extracts comprising these compounds also have produced different elegant shades on different types of fabrics and have been currently investigated as novel functional agents in the production of multifunctional textile surfaces such as deodorizing, antioxidant [5,7,8,29,31–34], antimicrobial [7,10,13–18,29,35], UV protection [15,18,32,34,36], and flame retardancy [13] properties [2].

*Rosa canina* grows naturally in a wide geography in the world including Central and Western Asia, Caucasus, Europe, Northwest Africa, northern and western parts of Iraq and Iran, northern Afghanistan, Pakistan, Kashmir, and the Commonwealth of Independent States [37]. Approximately 25% (27 species) of *Rosa canina*, of which 70–100 species are grown in the world, are grown in Turkey [38]. They are not very selective in terms of climate and soil requirements. Due to this feature, the plants can grow in almost every ecology, different soil types, and different altitudes. They are very resistant species to extreme climatic conditions. Since they loom between May and June, there is no risk of being damaged by late spring frosts, and they are extremely resistant to drought [37]. In addition, the fruits of *Rosa canina*, have a very long lifespan (30–40 years) and grow naturally without any requirement for chemical compounds or fertilizers [39].

Despite the variety of species, *Rosa canina* contains about 20 to 30 times more vitamin C than oranges. In addition to being a valuable mineral source, they are highly rich in phosphorus and potassium [40–42]. Therefore, *Rosa canina* fruits are widely used in the food and pharmaceutical industries. Many foods such as marmalade, jam, walnut sausage, nectar, and herbal tea are made from these fruits in Turkey [38,43,44]. In addition, they are added to probiotic drinks, fruit yogurts, and soups [45] and are also used in landscaping [46]. The fact that they can be used in textiles as well as in all these areas will further increase the production of the fruit and the breeding of the wild population. In the literature, as far as has been reported, there is only one study as a preliminary study on the use of *Rosa canina* fruits as a potential natural dyestuff in the textile industry [17].

Regarding the extraction method of *Rosa canina*, supercritical extraction [47], traditional extraction with deionized water with a Soxhlet device [17,48], traditional hot water infusion and boiling with water [49], and ultrasonic extraction [50,51] methods are available.

Since the components of plants and other natural ingredients such as polyphenols, flavonoids, and tannins are very sensitive to temperature, in recent years, Ultrasonic-assisted extraction (UAE) and modern heating technologies like IR and microwave [52,53] have been identified to enhance the effectiveness of the process. Extraction methods of *Rosa canina* differ, and some include boiling solvents (water: 100 °C, methanol: 64.7 °C, and ethanol: 78.4 °C) which can damage the active components.

There is no detailed and sensitive ultrasound-assisted extraction study of *Rosa canina*, which also uses three different solvents (water, ethanol, methanol, and water/methanol) with a temperature up to 60 °C. A total of 42 different phenolic compounds were found in *Rosa canina* by other studies [54,55]; it was important to develop an effective extraction

process to obtain a maximum amount of phenolic compounds without any damage or degradation and apply them to textile fabrics.

Therefore, in our previous study [56], a novel ultrasound-assisted extraction process for *Rosa canina* was developed as a possibility of *Rosa canina* extractions' use in the textile area as a natural functional agent for fabrics. For this purpose, different solvents such as distilled water, ethanol, and methanol were used in ultrasound-assisted extraction. Experiments were carried out at different temperatures (30, 45, and 60 °C) and times (30, 45 and 60 min). All the results were examined by UV-Vis spectrometry and optimum parameters were determined in terms of the amount of the active compounds obtained. Results showed that water as a solvent for *Rosa canina* extraction leads to the highest amount of active compounds, followed by methanol and ethanol solvents. As process parameters, 60 °C and 60 min were found to be ideal for the maximum extraction performance of *Rosa canina* via an ultrasound device [56]. Regarding its textile applications, *Rosa canina* extraction treatments slightly imparted UV protection properties to cotton fabrics. However, 50 g/L fruit concentration was not sufficient to achieve desirable Ultraviolet Protection Factor (UPF) values. Moreover, extraction with ethanol solvent was found to be the least effective in achieving active compounds, so it was eliminated in further steps [57].

In this recent study, initially, both fresh and dry *Rosa canina* fruits were extracted using an ultrasound device at 60 °C for 60 min in the solvents of water, methanol, and water/methanol mixture (50/50% *v/v*) with a fruit concentration of 200 g/L. All extractions were then analyzed by UV-Vis spectrometry as well as HPLC-UV analysis for in-detail chemical investigation. Secondly, the pre-mordanting process was carried out with chitosan solution via the pad–dry–cure method using a laboratory padder to investigate the possibility of increasing the efficiency of the dyeing and finishing process as well as the potential of chitosan to provide antibacterial properties for cotton fabrics. Chitosan was selected due to its status as one of the most plentiful, biodegradable, renewable, and non-toxic biopolymers found in nature. It is well suited for replacing certain synthetic polymers.

Chitosan, illustrated in Figure 1, is a polymer consisting of b-1, 4-linked glucosamine residues and is typically derived by deacetylating chitin from shrimp and other shells using a concentrated sodium hydroxide solution [2,58,59]. It is also a hydrophilic polymer [60] and attracts great interest in textile applications, from fabric pre-mordanting to finishing processes [61]. As it is a linear heterogeneous cationic polysaccharide with multiple properties, it has many properties such as antibacterial action, biodegradability, hydrophilicity, non-toxicity, biocompatibility, and adsorption properties, making it a widely used material for textile mordanting and finishing applications. Particularly, it is used in textile processes to add functionality, improve dyeability and physical characteristics, provide wrinkle resistance, and introduce bioactivity. The bactericidal efficacy of the chitosan biopolymer relies significantly on factors such as positive charge density, molecular weight, concentration, hydrophilic/hydrophobic properties, chelating capacity, and the physical state of the biopolymer [62,63]. Concerning the application of chitosan as a mordant in cotton fabrics, both cotton and chitosan share similar structures, leading to the linkage of their main molecular chains through Van der Waals forces. Additionally, cross-linking occurs through Schiff base formation between the reducing end groups of cellulose and the amino groups of chitosan [62]. Furthermore, from the perspective of sustainability, the European IPPC Bureau lists it in the BAT (Best Available Techniques) document as a good finishing aid that can replace less sustainable, conventional ones [64].

Thirdly, *Rosa canina* extractions were applied to untreated and pre-treated cotton fabrics via the exhaustion method at 60 °C for 60 min. All treated fabrics were analyzed by Fourier-transform infrared spectroscopy (FTIR) to investigate the chemical structure of the surfaces. Lastly, they were tested for their functionality by UV protection and antibacterial activity tests. Color measurements were also performed using a visible spectrophotometer in terms of CIELAB values (L*, a*, b*) and color strength (K/S) to analyze the color changes of fabrics and dyeability properties of the extractions. Finally, the wastewater (waste dye

solutions) was analyzed by UV-Vis spectrometry for the investigation of the potential transfer of active compounds to the fabrics.

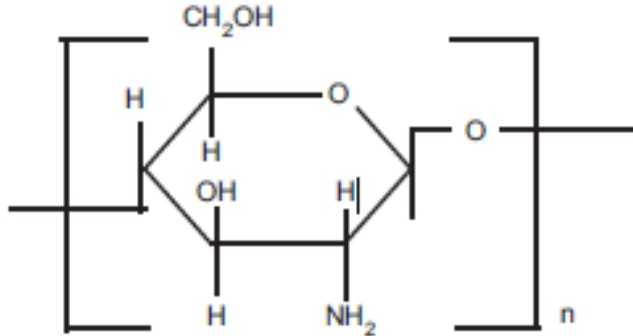

**Figure 1.** Chemical structure of chitosan.

## 2. Materials and Methods

### 2.1. Materials

*Rosa canina* samples were collected from the Konya and Gumushane regions in Turkey in October 2021. Both fresh and dry fruit versions were collected and stored at −18 °C. The maturity stage (ripeness) of the samples was classified by the color of the fruit. All samples used in the project were orange and red colored samples, which were characterized as fully ripe. In all experiments, fruit samples were used with their seeds and were mechanically broken into pieces to increase extraction efficiency. Yavuz Textile in Turkey supplied 100% cotton fabric (twill, 255 g/ready to dye).

All chemicals used in this study were of analytical reagent grade, and distilled water was used throughout. Methanol was provided from AppliChem ITW Reagents and was of >99.5% mass fraction purity. Chitosan, vanilic acid, caffeic acid, ferulic acid, gallic acid, p-couramic acid, and protocautehauic acid were supplied by Sigma-Aldrich, Darmstadt, Germany.

### 2.2. Preparation of Extracts

The extraction stage was performed using both dried and fresh *Rosa canina* samples with an ultrasound device. To optimize the extraction procedure, extraction process parameters were defined as solvent type (water, methanol, and mixture of water and methanol), fruit concentration (200 g/L), temperature (60 °C), and time (60 min).

Ultrasound-assisted extraction was conducted in an ultrasonic bath (P-Selecta Ultrasons H-D device, Barcelona-Spain, 40 kHz) at the selected temperature. *Rosa canina* samples and solvents were sealed in an Erlenmeyer flask and placed into the bath. The mixture was centrifuged for 60 min. After centrifugation, extract solutions were filtered through a filter paper, then bottled, covered with aluminum foil, and stored in the fridge (at around 4–5 °C) in a dark environment until chemical analysis. The pH values of extractions were also measured as 4 for dry fruit water extraction, 4–5 for dry fruit water/methanol extraction, 5 for both fresh fruit water and fresh fruit water/methanol extractions, and 6 for both dry fruit methanol and fresh fruit methanol extractions.

### 2.3. Analysis of Extract Solutions by UV-Vis Spectrophotometry

All extraction solutions were analyzed using UV-vis spectrophotometry (Hitachi UH5300, Ibaraki, Japan) in terms of the amount of active compounds. UV-vis graphs were overlapped and compared to identify the optimum extract solution. With UV-Vis spectrophotometry analysis, it is possible to make a comparison between extractions in terms of the amount and the variety of compounds. However, more complicated analytical methods such as HPLC-DAD/-ESI-MS and HPLC-UV-MS are required to identify the exact chemical constituents (carotenoids, types of phenolic compounds such as catechin, gallocatechol, quercetin, rosmarinic acid, apigenin, resveratrol, etc.) present in the extractions.

### 2.4. Analysis of Extract Solutions by HPLC-UV

#### 2.4.1. Preparations of Standard Solutions

For the preparation of the standard calibration curves, the stock solutions of vanillic acid, caffeic acid, ferulic acid, gallic acid, p-couramic acid, and protocatechuic acid were prepared in acetonitrile at a concentration of 500 mg/L. The concentrated solutions were then diluted with acetonitrile to obtain 1 mg/L. All solutions were filtered through a 0.45 μm PTFE membrane filter.

#### 2.4.2. Chemical Analysis by HPLC-UV

To identify active compounds in the extractions, 200 g/L *Rosa canina* extractions (both fresh and dry fruit) with water and methanol were analyzed through HPLC-UV using the chromatographic method (0.5 mL/min formic, acid 0.1 M: acetonitrile (85:15) isocratic, 40 °C, C18 column). The HPLC system utilized for this study included a Hitachi Chromaster with a pump (model 5110), UV/Vis detector (model 5410), oven (model 5310), and autosampler (5210 model). A C18 Machery-Nagel column (Nucleodur- $\pi$ 5 μm) served as the stationary phase, and elution was carried out with an isocratic flow of 0.5 mL/min using a mixture of 0.1 M formic acid and acetonitrile (85:15). The temperature of the oven was adjusted to 40 °C, and detection occurred at 275 nm. Mineralization analyses were conducted using a Shimadzu TOC-V instrument equipped with an ASI-V autosampler. HPLC graphs were overlapped to compare the amount of active compounds in the solutions.

### 2.5. Pretreatment Process

To investigate the possibility of increasing the efficiency of the finishing process, the pre-mordanting process with chitosan as a bio-mordant was carried out via the pad–dry–cure method. A solution containing 5 g/L of chitosan was prepared by adding 3 g/L of acetic acid to facilitate the dissolution of chitosan. The mixture was stirred magnetically for a duration of 24 h until a fully dissolved solution was achieved. Amounts of fabrics weighing 5 g were cut and put in an oven at 60 °C for 60 min to remove their moisture. Then, they were weighed again and noted. Each fabric then was immersed into chitosan solution and squeezed using a laboratory padder with a pick-up ratio of 90–95%. Wet fabrics were dried in an oven at 100 °C for 15 min, cured at 150 °C for 3 min, and left in laid form in laboratory conditions. After 24 h, all fabrics were put in an oven at 60 °C for 60 min to remove their moisture, weighed again, and noted to calculate add-ons.

### 2.6. Textile Dyeing & Finishing Process

For textile dyeing and finishing processes, 100% cotton chemically bleached fabric (Prepared to dye-255 g/m$^2$, 2/1 twill Z) was selected. The dyeing process was performed using a linitest device (Testherm 90S, Kraków, Poland) via the exhaustion method at 60 °C for 60 min. Fabric samples weighing 5 g were cut, and placed in vessels with 125 mL of extraction solutions of water, methanol, and a mixture of methanol and water (50–50% *v/v*) (the liquor ratio: 1:25). After the dyeing process, fabrics were taken out, squeezed, and left to dry at room temperature for at least 24 h. Waste solutions (usually called wastewater), which were dye solutions left in the steel vessel in the linitest device after the dyeing process, were filtered, and 100 mL of those were kept in the refrigerator for further analysis by UV-Vis spectrophotometry.

### 2.7. Characterization of Fabrics by FTIR

The surface chemistry and chemical composition of the untreated and treated fabrics was characterized by Fourier-transform infrared spectroscopy (FTIR) using an FT/IR-4700 (Palo Alto, CA, USA) type A from JASCO with ATR accessory, at a wavenumber range of 500–4000 cm$^{-1}$. In total, 16 spectra were recorded with a 4 cm$^{-1}$ resolution.

### 2.8. UV Protection of Fabrics

Untreated and *Rosa canina* extraction-treated fabrics at a concentration of 200 g/L were assessed for their Ultraviolet Protection Factor (UPF) properties using a methodology that involved the utilization of a UV lamp, a digital UV radiation detector, and an enclosed opaque box. The assessment system comprised a VL-6.C UV radiation lamp emitting at 312 nm for UVB and 365 nm for UVA radiation, a photoelectric sensor (Delta Ohm HD 2102.2) connected to a computer, and a light-proof box that envelops the entire apparatus to prevent interference from external lighting. The photoelectric sensor absorbs the entire radiation emitted by the UV lamp. The fabric is positioned above the UV lamp, and the entire setup is enclosed within an opaque box to eliminate external light interference. Test samples measuring 10 × 10 cm were employed to perfectly cover the detector area. Measurements were taken from three different positions on each fabric sample to derive an average result. UPF results were then calculated based on the average UVA and UVB values obtained from the measurements.

### 2.9. Antibacterial Activity of Fabrics

The antimicrobial efficacy of cotton fabrics was evaluated following the AATCC Test Method 100-2012 against both gram-positive and gram-negative bacteria. *Staphylococcus aureus* (AATCC 6538) was employed as the gram-positive bacterium, and *Escherichia coli* (AATCC 25922) was used as the gram-negative bacterium, as they are the major causes of cross-infection in hospitals.

### 2.10. Color Measurements of Fabrics

The color parameters of untreated and treated cotton fabric samples, which were folded twice, were measured following a standard procedure. Color parameters were assessed using CIELAB values (L*, a*, b*) and color strength (*K/S*) under standard illuminant D65 and a 10° observer, with specular radiation excluded. A Minolta CM-3600d (Ramsey, NJ, USA) visible spectrophotometer, equipped with SpectraMagic$^{TM}$ NX software, was employed for the evaluation. The Kubelka–Munk equation (Equation (1)) was applied to measure the relative color strength (expressed as *K/S* value) of various treated cotton fabrics through the light reflectance technique.

$$\frac{K}{S} = \frac{(1-R)^2}{2R} - \frac{\left(1-R^0\right)^2}{2R^0} \tag{1}$$

where *K* represents the absorption coefficient, *S* denotes the scattering coefficient, and *R* stands for reflectance. The overall color difference for cotton fabric samples was determined using the subsequent relationships (Equation (2)):

$$\text{Color difference}(\Delta E) = \sqrt{(\Delta L)2 + (\Delta a)2 + (\Delta b)2} \tag{2}$$

## 3. Results

### 3.1. UV-Vis Analysis of Extraction and Waste Solutions

The UV-Vis absorption spectroscopy of *Rosa canina* extractions with different solvents provides an amount of detail about their active compounds, which can be transferred into fabrics and their coloring capacity in the textile dyeing and finishing process. Moreover, UV-Vis absorption spectra of waste solutions also supply the amount of active compounds, which could not be transferred to fabrics. Therefore, it is essential to study the UV-Vis analysis of both extraction and wastewater and compare them in graphs. UV-Vis spectra of dry and fresh *Rosa canina* extractions are shown in Figure 2.

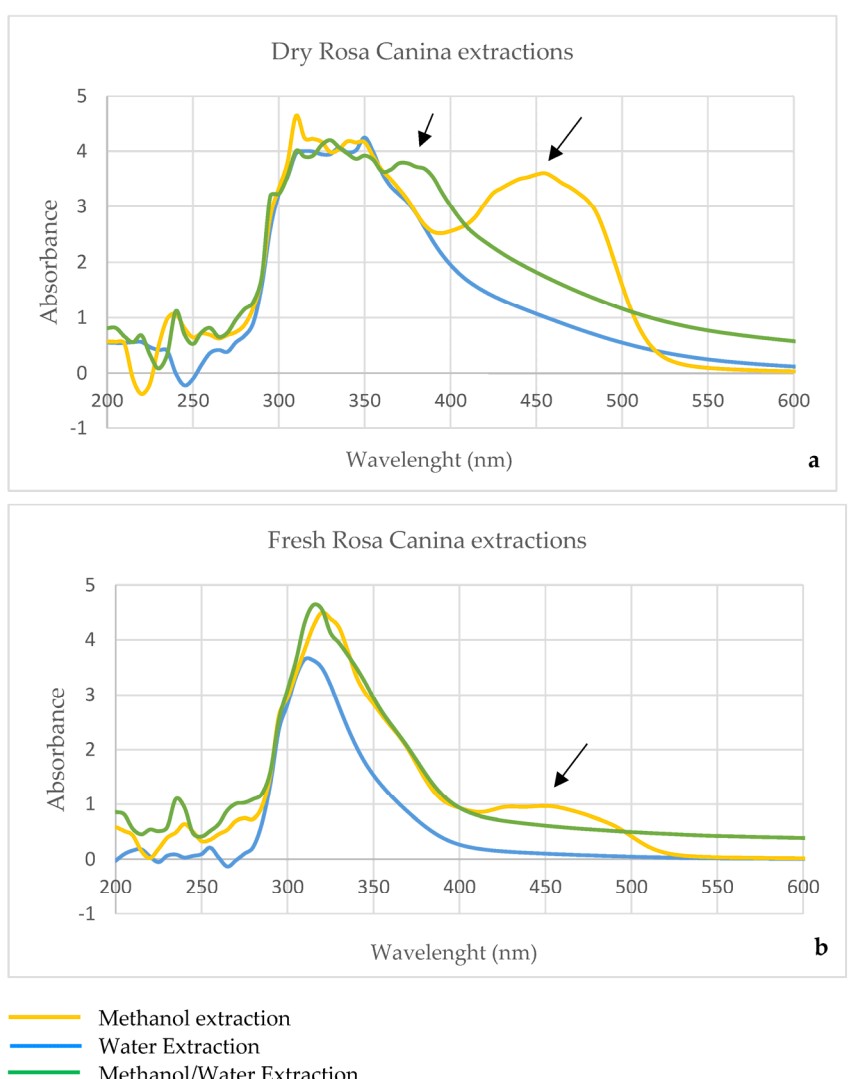

**Figure 2.** UV-vis spectra of dry and fresh *Rosa canina* extractions in different solvents.

As seen in Figure 2a, regarding dry *Rosa canina* extractions, all solutions show around three different absorption peaks with an average absorbance intensity of four at a wavelength of 300–350, which indicates the variety of bonds that the active compounds present in these extractions. All these dry fruit extractions probably have similar active compounds in similar amounts. Differently, water/methanol extraction has an extra absorption peak at 375 nm with an intensity of 3.7, which confirms that this bond can only occur in the presence of alcohol and water. The appearance of an extra peak suggests the potential presence of a compound or compounds that are soluble in the combined water and methanol solvent but not in water or methanol alone.

However, in the UV-Vis spectra of fresh *Rosa canina* extractions seen in Figure 2b, there is only one dominant absorption peak with an intensity value of 4.5 for methanol and methanol/water extractions and 3.5 for water extraction at a wavelength around 320 nm, which manifests a higher amount of one active compound in methanol and methanol/water solvents than in those of dry *Rosa canina*.

If all these active compounds in this region of 300–350 can be successfully transferred to cotton fabrics and provide some functional properties, dry *Rosa canina* extractions would have advantages over those with fresh *Rosa canina* with more varieties of bonds. Moreover, for all dry and fresh fruit extractions at wavelengths of 200–280 nm, the absorption intensity is less than one, which can play a less significant role in providing functionality to fabrics due to weak bonds.

When compared to solvents, in fresh *Rosa canina* extractions, methanol and methanol/water solvents seem to be more effective than water in extracting the dominant active compound, as seen at a wavelength of 320 nm. In addition, methanol extractions of both fresh and dry *Rosa canina* have another absorption peak in the visible region centered at a wavelength around 450 nm, which indicates some coloring features. This absorption peak is higher in dry fruit methanol extractions with an absorption intensity of 3.5 than in fresh fruit ones with a value of 1. Therefore, it can be said that the color features of methanol extractions are stronger in dry fruit extractions than in fresh fruit ones. On the contrary to dry fruit extraction, fresh fruit extraction of methanol/water does not demonstrate an extra peak in the visible region.

UV-Vis spectra of dry and fresh *Rosa canina* extractions in terms of solvents and their waste solutions are shown separately in Figures 3–5.

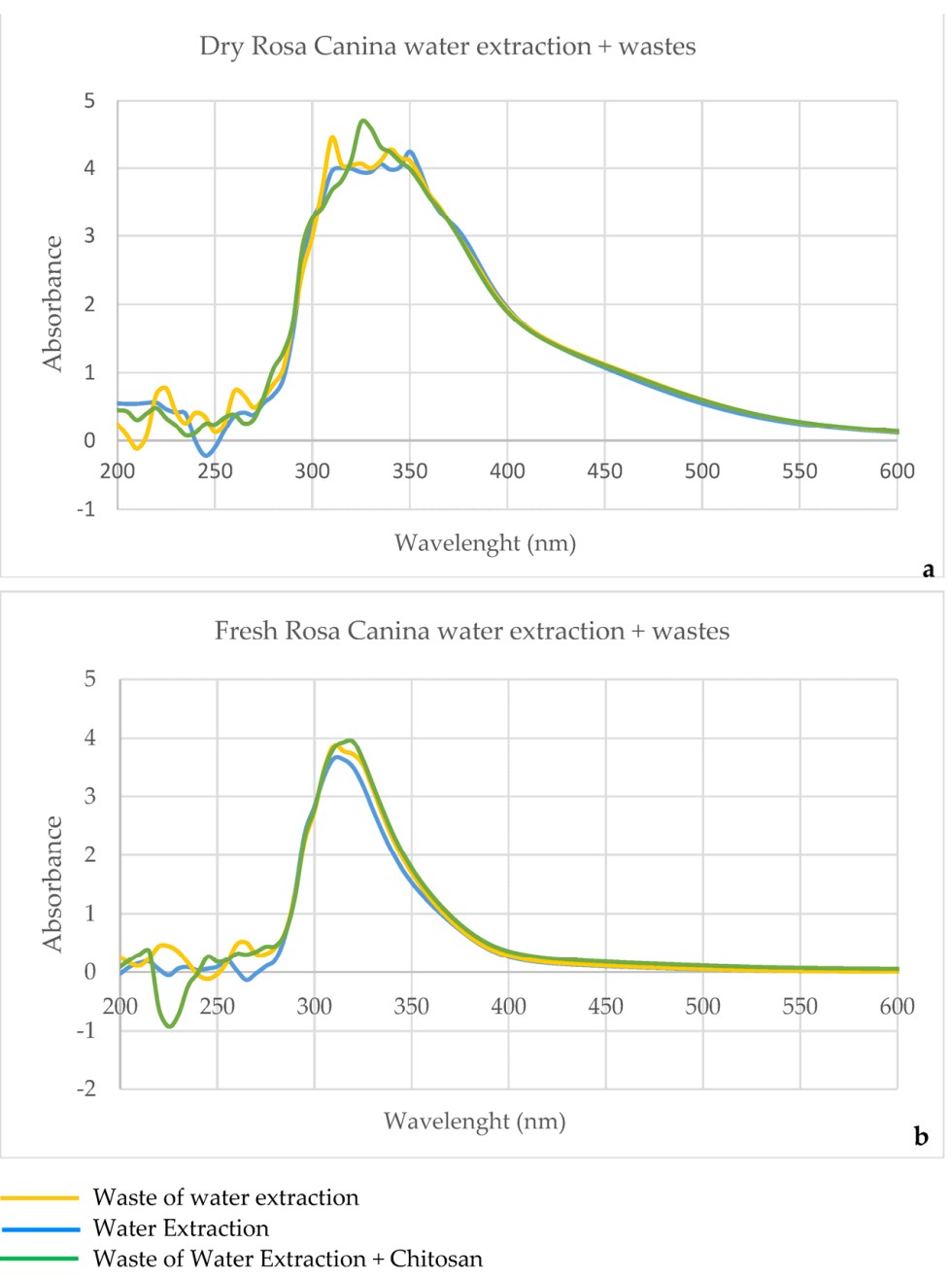

**Figure 3.** UV-vis graphs of *Rosa canina* extractions with water and waste solutions.

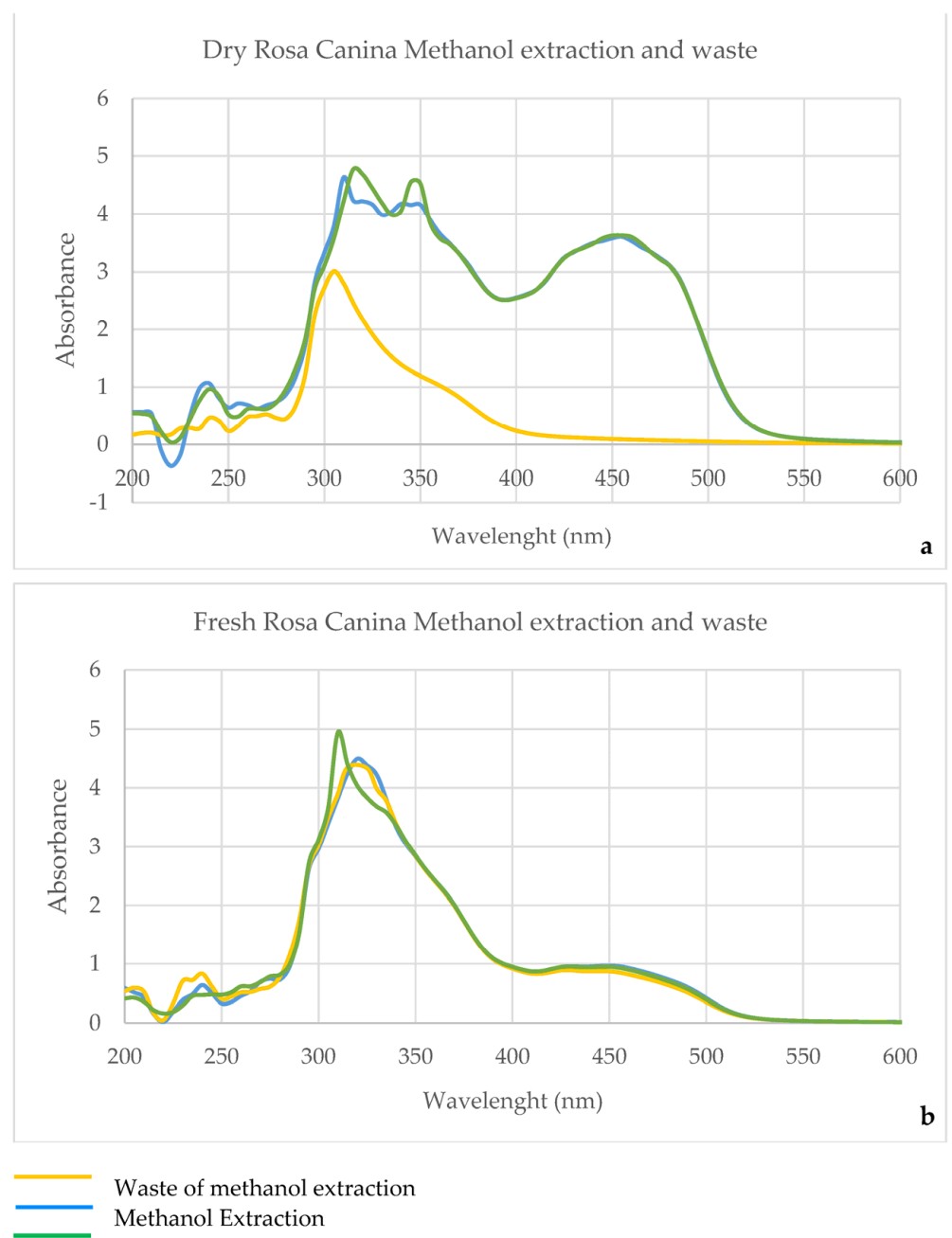

**Figure 4.** UV-Vis spectra of *Rosa canina* extractions with methanol and waste solutions.

It is seen in Figure 3a that the UV-Vis spectra of dry fruit water waste solution show a very similar trend to water extraction except for the absorption peak at 310 nm, indicating that waste extractions still have similar bonds to extractions. Differently, waste extraction with chitosan demonstrates only one absorption peak at 330 nm with an intensity of 4.5, which could indicate that the active compounds observed at wavelengths of 315, 340, and 345 nm in dry fruit water extractions have been transferred to cotton fabric in the presence of chitosan. It also indicates that, on the other hand, a new bond occurs between one active compound in the extraction and chitosan at a wavelength of 330 nm. However, chitosan, which shows a peak at 295 nm in its UV-Vis spectrum [65], cannot be seen in any of the spectra in Figure 3 that demonstrate chitosan has been conveniently bonded to the cotton fiber.

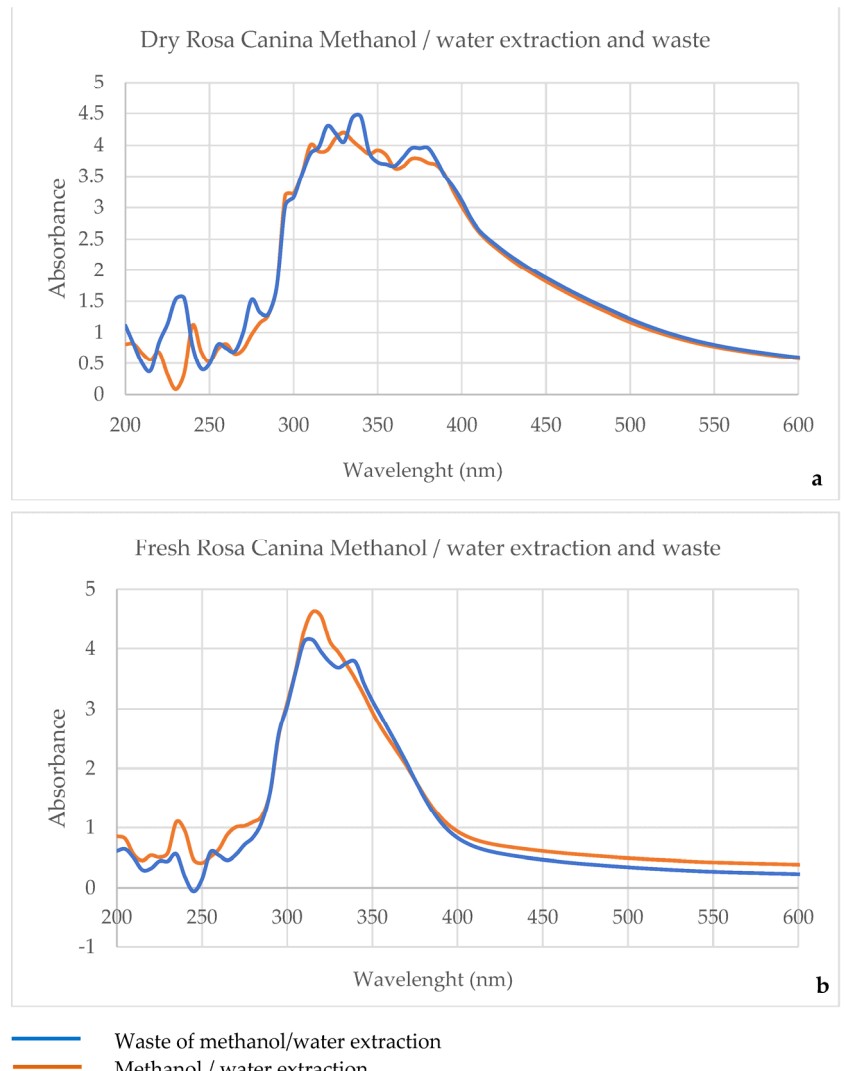

**Figure 5.** UV-vis spectra of *Rosa canina* extractions with methanol/water and waste solutions.

In UV-Vis spectra of fresh *Rosa canina* seen in Figure 3b, extractions and waste solutions also demonstrated a very similar trend, showing absorption peaks at around a 320 nm wavelength, with a maximum absorption value of four, which indicates that extraction and waste solutions have the same active compounds. In waste solutions, the bonds are slightly stronger than those in the original extraction due to the presence of unreacted extract components, possible impurities from cotton fabric, and residual of chitosan.

In the UV-Vis spectra of dry *Rosa canina* methanol extractions and wastes shown in Figure 4a, the waste solution has only one absorption peak with a lower intensity, which is a desirable result in contrast to methanol extraction itself and waste solution with chitosan. In the spectrum of waste solution with chitosan, sharper peaks with higher absorption intensities were observed compared to methanol extraction itself, which manifests that chitosan probably has bonded with active compounds in the extract solution. The same trend can be observed in the UV-Vis spectra in Figure 4b as the waste solution including chitosan has a sharp peak with an absorption intensity of five at 310 nm, while the spectra of methanol extraction and waste solution have almost the same trend.

According to the UV-Vis spectra in Figure 5, similar trends were observed for both original extractions and their waste solutions as they show similar absorption peaks with intensity between 3.5 and 4.5 at 300–400 nm wavelengths. In Figure 5b, only one sharp adsorption peak with an intensity of 4.5 was observed for extraction at 320 nm; however,

the waste solution shows two different bonds with intensities of 3.7 and 4.1 at 340 nm and 315 nm, respectively. That result indicates that the waste solution of fresh *Rosa canina* methanol/water extraction still has some unreacted active compounds, methanol, water, and also impurities of cotton fabric, which can lead to new bonds with each other.

### 3.2. HPLC-UV Analysis of Rosa canina Extractions

HPLC-UV graphs of water and methanol extraction of *Rosa canina* (200 g/L) are seen in Figure 6.

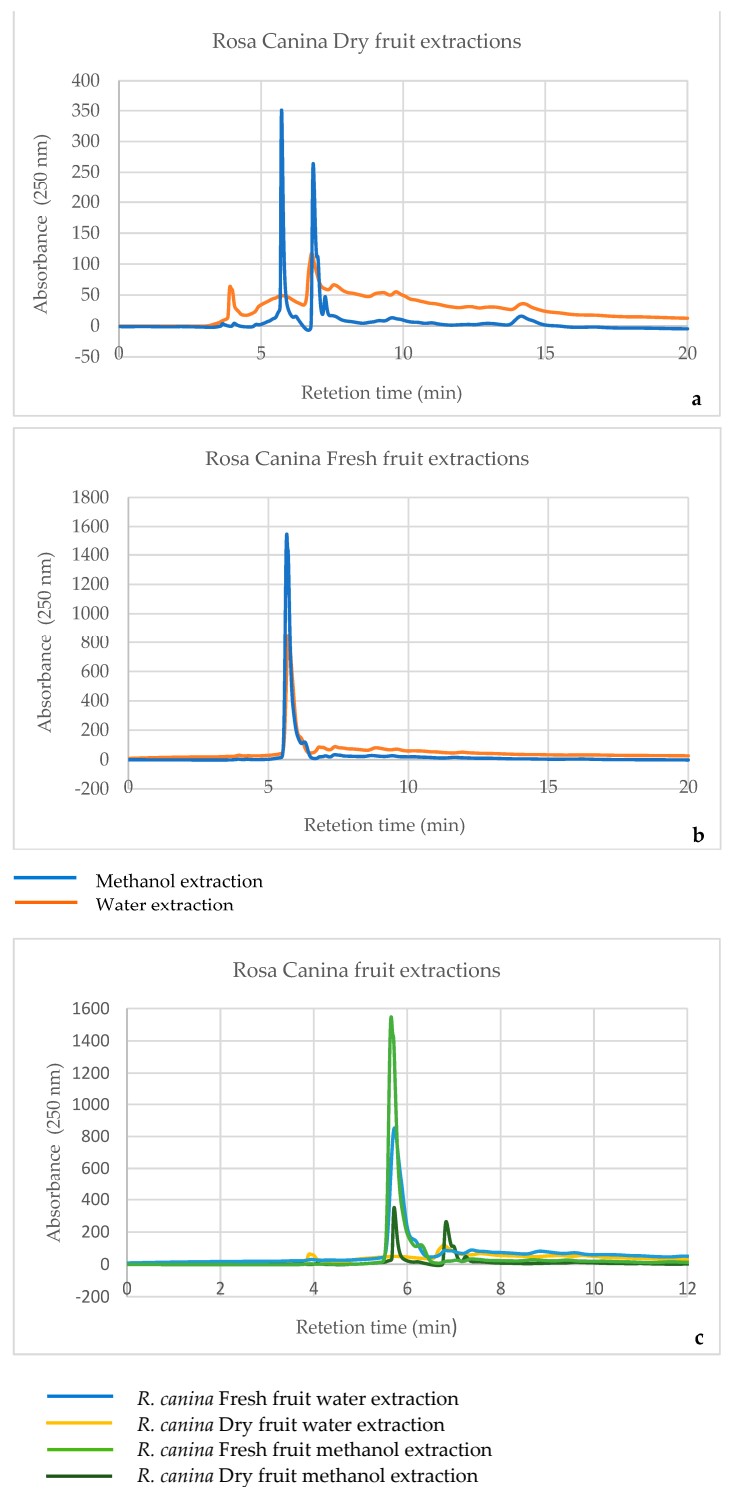

**Figure 6.** HPLC-UV spectra of *Rosa canina* extractions with water and methanol.

In the HPLC-UV spectra of dry *Rosa canina* extractions in Figure 6a, they have two dominant absorption peaks at 5.72 and 6.83 retention times, which indicates that there are definitely two dominant active compounds in the extractions. On the other hand, in the HPLC-UV graphs of fresh *Rosa canina* extractions in Figure 6b, these two peaks obviously show higher intensities ($\lambda$), which manifests that these two components are present in higher amounts in fresh fruit extractions.

Apart from these two components, as it is seen in Figure 6c, *Rosa canina* water extractions show similar peaks with those of methanol extractions at retention times of 3.91, 9.29, 9.74, and 14.22 with higher absorption intensities, which demonstrates the presence of other active compounds in all extraction solutions with higher amounts in water extractions. Therefore, water extractions have advantages over methanol extractions with a variety of components in higher amounts.

Moreover, in both Figure 6a,b, in the HPLC-UV spectra of fresh *Rosa canina* extractions with both water and methanol, methanol extraction seems to be more efficient, with a plausible advantage of concentrating the analyte by solvent evaporation. The main extracted compound, evidently seen at a retention time of 5.7, seems to be a very polar molecule.

In addition, the retention times of stock solutions of vanillic acid, caffeic acid, ferulic acid, gallic acid, p-couramic acid, and protocatechuic acid and extractions are presented in Tables 1 and 2.

**Table 1.** Retention times of stock solutions used in HPLC-UV.

| Standard 1 mg/L | Retention Time (min) | $\lambda_{max}$ |
|---|---|---|
| Gallic acid | 8.77 | 3.39 |
| Protocatechuic acid | 12.97 | 9.85 |
| Vanillic acid | 21.13 | 8.15 |
| Caffeic acid | 24.86 | 3.88 |
| p-couramic acid | 42.95 | 0.88 |
| Ferulic acid | 45.42 | 3.03 |

**Table 2.** Retention times of extract solutions detected in HPLC-UV.

| *Rosa canina* fresh fruit methanol extraction | | *Rosa canina* fresh fruit water extraction | |
|---|---|---|---|
| **Retention time (min)** | $\lambda_{max}$ | **Retention time (min)** | $\lambda_{max}$ |
| **5.65** | **1548.13** | **5.72** | **852.64** |
| **6.29** | **120.68** | **3.95** | **30.14** |
| 7.02 | 25.74 | 6.93 | 83.6 |
| 7.37 | 33.71 | 7.39 | 88.58 |
| 8.69 | 27.98 | 8.85 | 81.12 |
| 9.43 | 26.56 | 9.57 | 71.49 |
| 11.65 | 15.62 | 11.92 | 50.54 |
| *Rosa canina* dry fruit methanol extraction | | *Rosa canina* dry fruit water extraction | |
| **Retention time (min)** | $\lambda_{max}$ | **Retention time (min)** | $\lambda_{max}$ |
| **3.7** | **1.99** | **3.91** | **64.03** |
| **4.08** | **3.96** | **5.71** | **49.41** |
| 5.72 | 350.92 | 6.78 | 117.7 |
| 6.83 | 264.02 | 7.57 | 66.46 |
| 7.25 | 47.37 | 9.29 | 54.04 |
| 9.62 | 13.23 | 9.74 | 55.1 |
| 14.15 | 15.84 | 14.22 | 36 |

According to Tables 1 and 2, none of the injected standards (vanillic acid, gallic acid, caffeic acid, ferulic acid, p-couramic acid, and protocatechuic acid) are in significant concentrations within the extraction samples. In HPLC-UV spectra, there were no vibration peaks at retention times of 12.97 (Protocatechuic acid), 21.13 (vanillic acid), 24.86 (Caffeic

acid), 42.95 (p-couramic acid), and 45.42 (Ferulic acid). It can be said that with a retention of 8.8, only gallic acid has been found in fresh *Rosa canina* extractions with low amounts (absorbance in water ext.: 81.12, in methanol ext.: 27.98).

### 3.3. FTIR Results

FTIR spectra of untreated cotton fabric, chitosan-treated fabric, and *Rosa canina* (R.C) extraction-treated fabrics are shown in Figure 7.

In the spectra of untreated and all treated cotton fabrics (Figure 7), the major vibration absorption peaks of cotton (cellulose) can be easily found. Detailed examination of Figure 7a reveals absorption bands at 3341 and 3283 cm$^{-1}$, corresponding to hydroxyl group stretching. Bands observed at 2905 cm$^{-1}$ and 1372 cm$^{-1}$ are attributed to stretching and deformation vibrations of the C-H group within the glucose unit. Characteristic peaks associated with CH wagging occur at 1430 and 1318 cm$^{-1}$. Signals at 1055 and 1031 cm$^{-1}$ are assigned to the -C-O- group of secondary alcohols and ethers functions present in the cellulose chain backbone [61,66].

In the case of chitosan-treated cotton fabrics, in the FTIR spectra, functional groups in chitosan show similar vibration absorption peaks and are mostly overlapped or shadowed by cellulose bands due to its cellulose-like structure as a polysaccharide and the low amount of use. As depicted in Figure 7a, a strong band within the 3283–3341 cm$^{-1}$ range corresponds to stretching of N-H and O-H, along with intramolecular hydrogen bonds. The absorption bands at approximately 2905 can be ascribed to symmetric stretching of C-H. The existence of residual N-acetyl groups is verified by the bands at approximately 1648 cm$^{-1}$ (indicating C=O stretching of amide I) and 1318 cm$^{-1}$ (representing C-N stretching of amide III). Bands around 1430 and 1372 cm$^{-1}$ confirm CH$_2$ bending and CH$_3$ symmetrical deformations, respectively. The absorption band at 1162 cm$^{-1}$ is attributed to the asymmetric stretching of the C-O-C bridge, while the bands at 1055 and 1031 cm$^{-1}$ correspond to C-O stretching. Notably, all these bands align with spectra observed in chitosan samples reported by others [58,61,67–69].

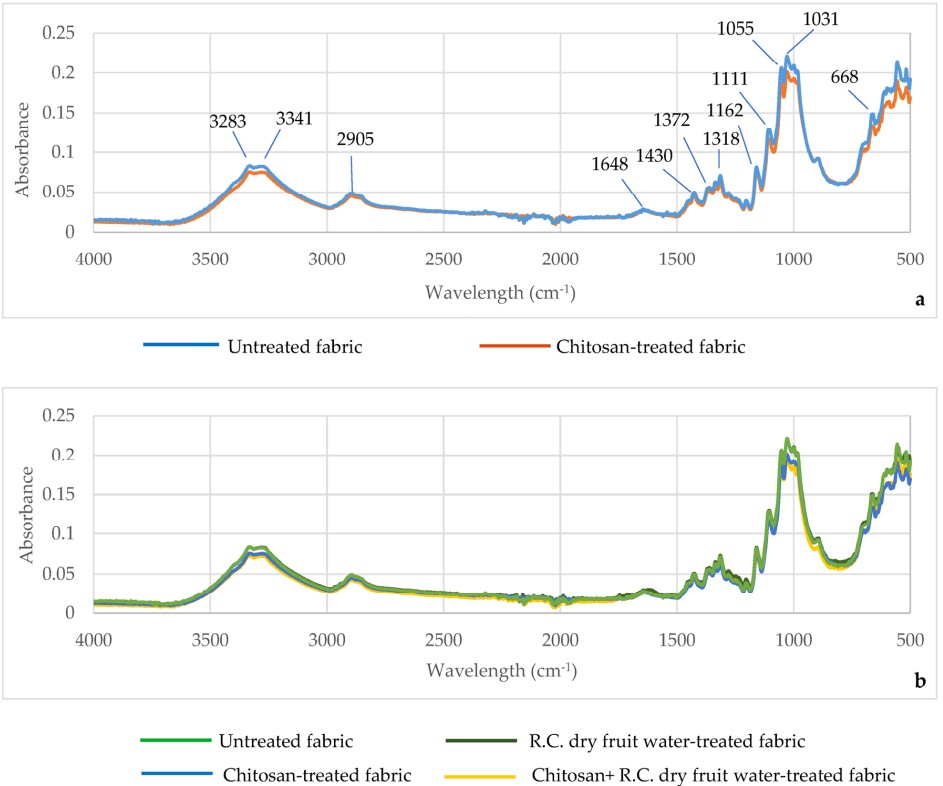

**Figure 7.** *Cont.*

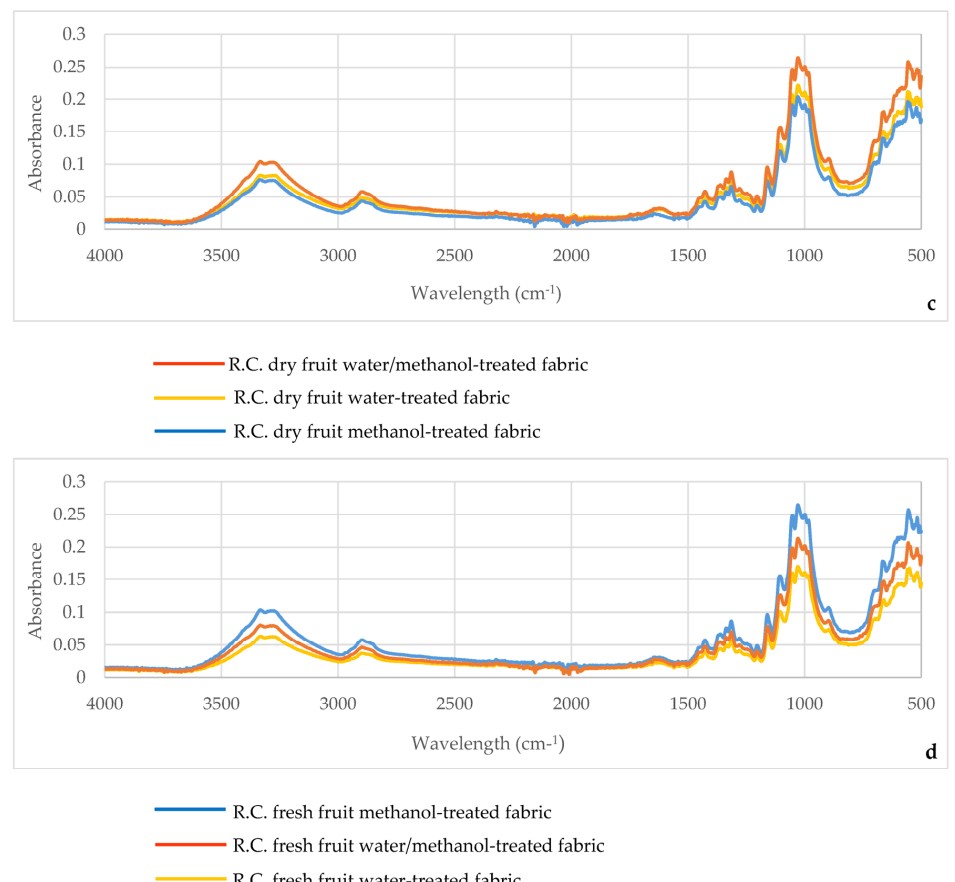

**Figure 7.** FTIR spectra of untreated, chitosan-treated, and *Rosa canina* extractions-treated cotton fabrics ((**a**) untreated and chitosan-treated, (**b**) untreated, chitosan-treated, *Rosa canina* dry fruit water-treated and Chitosan + *Rosa canina* dry fruit water-treated, (**c**) *Rosa canina* dry fruit water-treated, *Rosa canina* dry fruit water/methanol-treated, *Rosa canina* dry fruit methanol-treated, (**d**) *Rosa canina* fresh fruit water-treated, *Rosa canina* fresh fruit water/methanol-treated, *Rosa canina* fresh fruit methanol-treated).

When compared to absorbance intensities, as seen from the FTIR spectra in Figure 7a, untreated cotton fabric has higher absorbance intensities than the chitosan-treated ones. From Figure 7b, R.C. dry fruit water extraction-treated fabrics also show higher intensities than those of chitosan+ R.C. dry fruit water-treated fabrics. In Figure 7c, the FTIR spectra of R.C. dry fruit extraction-treated fabrics are seen in terms of different solvents. Methanol as a solvent has a lower effect than water only and water/methanol mixtures on absorption intensities. A water/methanol mixture as a solvent contributes higher absorbance peaks on the FTIR spectrum. In the case of R.C. fresh fruit extraction-treated fabrics, in Figure 7d, in contrast to dry fruit FTIR graphs, methanol extraction-treated fabrics have higher absorbance values. In general, in the FTIR spectra of untreated cotton fabric and the treated fabrics, the only difference is observed in the absorbance values, which indicates the presence of some additional compounds transferred to the fabrics. However, no extra peaks are observed.

### 3.4. Color Measurement Results

Colorimetric analysis of untreated and treated cotton fabrics, as presented in Table 3, provides valuable insights into the impact of various treatments on the color properties of the fabrics. The CIELAB color space parameters, including L*, a*, and b*, along with additional color-related (DE*ab, K/S) values, offer a comprehensive understanding of the treated fabric samples in comparison to the untreated reference.

**Table 3.** L*, a*, b*, DE*ab, and K/S values of untreated and treated cotton fabrics.

| No | Fabric Sample | L* | a* | b* | DE*ab | K/S (400 nm) | K/S (460 nm) | Color of Fabric |
|----|---------------|-----|------|------|-------|--------------|--------------|-----------------|
| 0 | untreated | 91.91 | −0.221 | 1.9386 | | 0.0459 | 0.0279 | |
| 1 | Chitosan+ R.C. dry fruit water ext. treated | 83.25 | 2.8814 | 12.935 | 14.341 | 0.5058 | 0.2314 | |
| 2 | Chitosan+ R.C. dry fruit methanol ext. treated | 85.96 | 3.7006 | 15.071 | 14.956 | 0.3052 | 0.2015 | |
| 3 | Chitosan treated (only) | 91.75 | −0.286 | 2.5508 | 0.7776 | 0.0548 | 0.0312 | |
| 4 | R.C. dry fruit water ext. treated | 86.25 | 2.1663 | 10.998 | 10.95 | 0.3693 | 0.1539 | |
| 5 | R.C. dry fruit methanol ext. treated | 87.70 | 3.5903 | 13.005 | 12.446 | 0.1815 | 0.1759 | |
| 6 | R.C. dry fruit water/methanol ext. treated | 86.12 | 2.0148 | 12.452 | 12.211 | 0.3708 | 0.1765 | |
| 7 | R.C. fresh fruit water ext. treated | 89.54 | 0.3429 | 6.6867 | 5.3536 | 0.198 | 0.0754 | |
| 8 | R.C. fresh fruit methanol ext. treated | 88.03 | 2.6744 | 9.7492 | 9.2075 | 0.2065 | 0.1159 | |
| 9 | R.C. fresh fruit water/methanol ext. treated | 89.14 | 0.9908 | 7.9718 | 6.758 | 0.194 | 0.0924 | |
| 10 | Chitosan+ R.C. fresh fruit water ext. treated | 86.76 | 0.5698 | 10.801 | 10.298 | 0.3483 | 0.1398 | |
| 11 | Chitosan+ R.C. fresh fruit methanol ext. treated | 87.81 | 2.4116 | 12.743 | 11.862 | 0.2792 | 0.1477 | |

The untreated cotton fabric (Sample 0) serves as the baseline reference. It exhibits a lightness (L*) value of 91.91, indicating a relatively light appearance. A negative a* value (−0.221) suggests a greenish tint, while a positive b* value (19.386) indicates a yellowish hue. Notably, a low DE*ab value implies minimal perceptible color difference from the reference. Regarding the impact of chitosan, Sample 3, treated with chitosan alone, demonstrates a subtle impact on color properties compared to chitosan combined with R.C. extracts. Samples 1 and 2, treated with chitosan in combination with *Rosa canina* (R.C.) dry fruit extracts using water and methanol, respectively, show significant alterations in color. Sample 1 exhibits a substantial decrease in lightness (L* = 83.25) and a considerable increase in chromaticity (a* and b* values) compared to the untreated fabric. Sample 2, treated with chitosan and R.C. dry fruit methanol extract, demonstrates a similar trend with enhanced color intensity and altered hue. As an effect of R.C. dry fruit extracts, Samples 4, 5, and 6, treated with R.C. dry fruit extracts using water, methanol, and a combination of both, respectively, showcase varying degrees of color modification. The choice of extraction solvent influences the resulting color, with methanol extraction (Sample 5) leading to a more pronounced change in color attributes. For comparison with fresh fruit extracts, Samples 7 to 11, treated with R.C. fresh fruit extracts with different solvents and in combination with chitosan, exhibit distinctive color profiles. The water/methanol extract combination

(Sample 9) showcases noteworthy alterations, suggesting a potential synergy between the solvents in extracting color-related compounds.

In summary, the color measurement results underscore the sensitivity of cotton fabric to various treatments. Chitosan and *Rosa canina* extracts, whether from dry or fresh fruits, introduce distinct color variations, offering a spectrum of aesthetic possibilities for tailored textile applications.

In addition to Table 3, color strength (K/S) curves of untreated and treated cotton fabrics are also shown in Figure 8. Color strength, as quantified by the K/S values, serves as a crucial indicator of a fabric's ability to absorb and reflect light, providing insights into the depth and intensity of color. As seen in Table 3 and Figure 8, the untreated cotton fabric exhibits a K/S value of 0.0459 at 400 nm and 0.0279 at 460 nm. These baseline values represent the inherent color strength of the untreated fabric, acting as a reference for assessing the impact of subsequent treatments. Fabrics treated solely with chitosan (Sample 3) show an increased K/S value at both 400 nm (0.7776) and 460 nm (0.0548). The rise in K/S values indicates enhanced color strength, likely attributed to the interaction of chitosan with fabric fibers, altering their optical properties. Fabrics treated with R.C. dry fruit extracts exhibit variations in K/S values depending on the extraction method. Water extract-treated fabric (Sample 4) demonstrates a significant increase in K/S values, indicating a boost in color strength. Methanol extract-treated fabric (Sample 5) shows a nuanced change in K/S values, suggesting alterations in color intensity. Fabrics treated with fresh fruit extracts (Samples 7–11) showcase distinctive K/S value patterns. Water and methanol extracts (Samples 7 and 8) lead to notable increases in color strength, particularly at 400 nm. Water/methanol combination treatments (Samples 9 and 11) exhibit unique K/S profiles, emphasizing the synergistic effects of extraction methods. Combining chitosan with fresh fruit extracts (Samples 10 and 11) results in K/S values that reflect a harmonized impact on color strength. These samples demonstrate a balance between the color-enhancing properties of chitosan and the unique color profiles introduced by *Rosa canina* extracts.

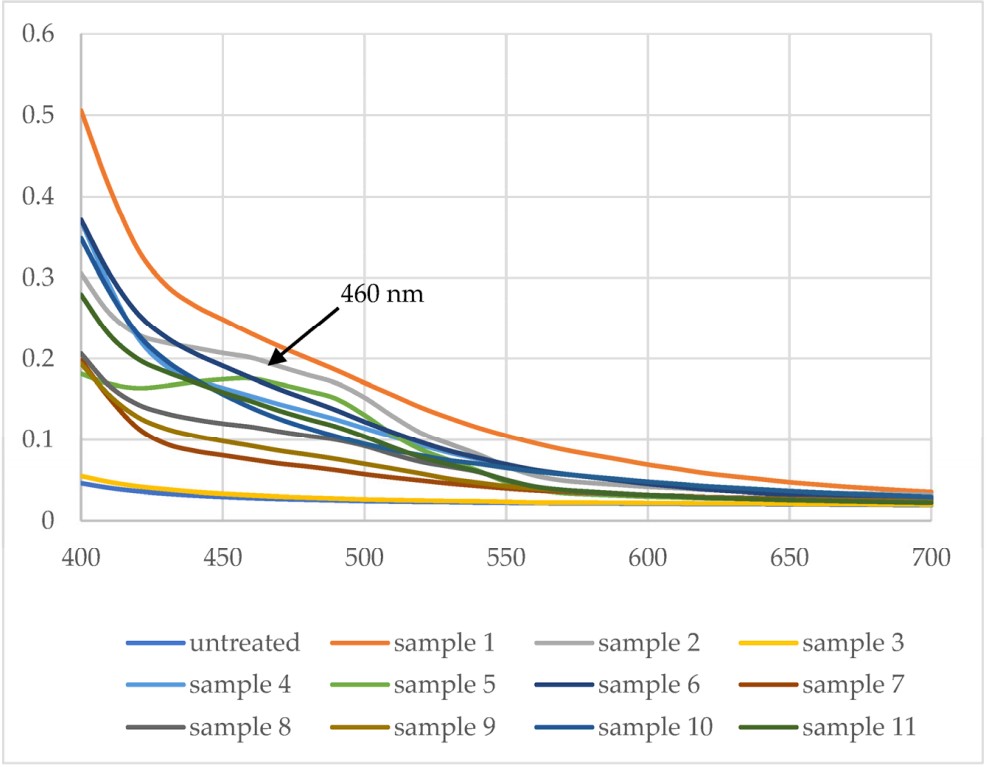

**Figure 8.** K/S curves of untreated and chitosan and R.C. extraction-treated cotton fabrics.

In summary, R.C. dry fruit water extractions lead to higher K/S values than those of fresh fruit extractions and methanol solvent extractions. It can be explained that some active compounds which only occurred in water extractions play a significant role in the color strength of fabrics. Most studies in the literature suggest that achieving a deeper shade in dyed fabric enhances its UV protection [70–72]. Hence, textile finishing with dry fruit extractions is expected to give more desirable UPF results for cotton fabrics due to higher K/S values (color depth).

Samples mordanted with chitosan show higher K/S than samples that are not mordanted. When L* is compared, L* decreases for samples with chitosan treatment which means the sample is darker. This suggests that chitosan acts as a binder between the fiber and the active compounds from fruit extracts. Thus, it is suggested that there is a $H_2$ bond from the hydroxyl groups (-OH) in the polyphenols in the active ingredients from the extraction and the ammonia ones (-NH$_3$) from chitosan. Thus, the concentration of active compounds is increased in the fiber, and consequently, the color strength is increased.

*3.5. UV Protection Results and Antibacterial Activities*

The UPF values, antibacterial activity results, and total add-on values of untreated and treated cotton fabric samples are shown in Table 4.

**Table 4.** UV Protection and antibacterial results of untreated and treated cotton fabrics.

| No | Fabric Samples | UV Protection (UPF) | Staphylococcus Aureus | *Escherichia coli* | Total Add-on (%) |
|---|---|---|---|---|---|
| 0 | untreated | 0.86 | no reduction | no reduction | (-) |
| 1 | Chitosan+ R.C. (dry) water ext. treated | 37.57 | 99.9% | no reduction | 7.37 |
| 2 | Chitosan+ R.C. (dry) methanol ext. treated | 16.36 | 65.0% | no reduction | 5.85 |
| 3 | Chitosan treated (only) | 1.36 | no reduction | no reduction | 0.94 |
| 4 | R.C. (dry) water ext. treated | 49.57 | no reduction | no reduction | 4.71 |
| 5 | R.C. (dry) methanol ext. treated | 12.64 | no reduction | no reduction | 1.32 |
| 6 | R.C. (dry) water/methanol ext. treated | 25.83 | no reduction | no reduction | 7.3 |
| 7 | R.C. (fresh) water ext. treated | 35.13 | no reduction | no reduction | 2.51 |
| 8 | R.C. (fresh) methanol ext. treated | 14.10 | no reduction | no reduction | 3.82 |
| 9 | R.C. (fresh) water/methanol ext. treated | 24.71 | no reduction | no reduction | 3.78 |
| 10 | Chitosan+ R.C. (fresh) water ext. treated | 34.64 | no reduction | no reduction | 5.22 |
| 11 | Chitosan+ R.C. (fresh) methanol ext. treated | 15.78 | no reduction | no reduction | 4.63 |

As seen from the UPF results in Table 4, in the first column, similar UPF values were obtained by untreated cotton fabric (Sample 0) and chitosan-treated cotton fabric (Sample 3) with values of 0.86 and 1.36, respectively, which indicates that chitosan treatment has a slightly positive effect on the UV protection of fabrics due to its low add-on (0.94%). In R.C. dry fruit extraction-treated samples, extractions with water seem to be more effective at providing UV protection, as R.C. dry fruit water extraction treatment (Sample 4)

provides the highest UPF (49.57), followed by chitosan+ R.C. dry fruit water extraction treatment (Sample 1: UPF of 37.57) and R.C. dry fruit water/methanol extraction treatment (Sample 6: UPF of 25.83). Among R.C. fresh fruit extraction-treated samples, the trend is the same: water extractions give more promising UPF, as R.C. fresh water extraction treatment (Sample 7) provides the highest UPF (35.16), followed by chitosan+ R.C. fresh fruit water extraction treatment (Sample 10: UPF of 34.64) and R.C. fresh fruit water/methanol extraction treatment (Sample 9: UPF of 24.71). In general, using dry fruits and water as a solvent for the extraction process of *Rosa canina* benefits over the fresh fruits and methanol or water/methanol mixtures in terms of the UV protection property of cotton fabrics as confirmed by higher color strengths (K/S values) on the color measurements.

In terms of the antibacterial activity of samples, chitosan treatment alone (Sample 3) shows a minor reduction in bacterial counts, and chitosan and *Rosa canina* extract treatments introduce varying degrees of antimicrobial effectiveness. The inclusion of *Rosa canina* extracts, especially in combination with chitosan, results in significantly enhanced antimicrobial activity against both Staphylococcus aureus and Escherichia coli. Samples 1 and 2, treated with a combination of chitosan and *Rosa canina* (R.C.) dry fruit extracts, demonstrated significant antibacterial efficacy against Staphylococcus aureus, achieving reductions of 99.9% and 65%, respectively. This observation suggests that the inclusion of chitosan, combined with R.C. dry fruit extractions, contributes to notable antibacterial properties of cotton fabrics. It can be explained that the pH of dry fruit water extractions is more acidic (pH:4) compared to dry fruit water/methanol extractions (pH:4–5), fresh fruit water and fresh fruit water/methanol extractions (pH:5), and dry fruit methanol and fresh fruit methanol extractions (pH:6). Because, the antimicrobial activity of chitosan, an amphoteric biopolymer, is pH dependent, with maximum efficacy achieved in acidic conditions, this is attributed to the improved solubility of chitosan in an acidic environment, forming a polycation [62].

Since, chitosan is a nontoxic, biodegradable, and biocompatible natural polymer with antimicrobial activity, the use of chitosan in antibacterial activities for textiles has been investigated for many years by several researchers [72–79]. It is a highly effective material in preventing bacterial growth and is increasingly becoming a standard finish for textile products. It is known that the external surface of most microbes is negatively charged, while chitosan is positively charged, and as a result, the binding capacity of chitosan is an important property and it is also interesting for biomedical applications as well as medical textiles [58].

Regarding add-ons (%), fabrics treated with *Rosa canina* dry fruit extractions have more add-ons than those treated with fresh fruit extractions. Maximum add-on values (7.37 and 7.30) were obtained by chitosan + *Rosa canina* dry fruit water extraction-treated (Sample 1) and *Rosa canina* dry fruit water/methanol extraction-treated (Sample 6) fabrics.

In summary, the synergistic effect of chitosan and R.C. dry fruit extracts not only enhances antibacterial performance against Staphylococcus aureus but also provides a probable source of active compounds contributing to UV protection properties in treated cotton fabrics. In addition, the effectiveness of the dry fruit extracts indicates that they likely contain a substantial quantity of active compounds responsible for conferring UV protection on the fabric. In contrast, fabrics treated with fresh fruit extracts showed comparatively lower antibacterial activity, implying a potential difference in the composition of active compounds between dry and fresh fruit extracts.

### 3.6. Scaling up Ultrasound-Assisted Rosa canina Extraction for Industrial Applications

While this study highlights the promising potential of ultrasound-assisted extraction (UAE) of *Rosa canina* for various applications in the textile industry, scaling up this process for industrial applications may encounter certain limitations and challenges in terms of equipment cost and maintenance, energy consumption, uniformity and consistency, process optimization, regulatory compliance, supply chain considerations, product stability and shelf life, economic viability, and waste management.

Equipment Cost and Maintenance: High-powered ultrasound equipment suitable for industrial-scale applications can be expensive to acquire and maintain. Regular maintenance and potential wear and tear of industrial-scale ultrasound equipment may pose challenges in terms of operational costs.

Energy Consumption: Scaling up the ultrasound-assisted extraction process may require higher energy inputs, contributing to increased operational costs and potential environmental concerns. Optimizing energy efficiency becomes crucial to ensure the economic viability of large-scale production.

Uniformity and Consistency: Achieving uniform extraction across a large-scale process can be challenging, leading to variations in the quality and quantity of extracted compounds. Ensuring consistent and reproducible results on an industrial scale requires meticulous control over various parameters, such as temperature, solvent flow, and ultrasound intensity.

Process Optimization: Optimizing extraction parameters, such as temperature, duration, and solvent to material ratio, becomes more complex on a larger scale. Extensive research and development are required to fine-tune these parameters for optimal results in an industrial setting.

Regulatory Compliance: Meeting regulatory standards for industrial-scale production involves rigorous testing and compliance checks for the extracted compounds. Ensuring that the extraction process meets industry and safety standards is essential for the acceptance of the final products in various applications.

Supply Chain Considerations: Sourcing an adequate and consistent supply of *Rosa canina* raw material for large-scale extraction can be challenging, potentially impacting the overall production chain. The dependence on seasonal variations and geographical factors requires strategic planning to maintain a stable supply.

Product Stability and Shelf Life: The stability of the extracted compounds and the shelf life of the final products become more critical considerations in industrial applications. Ensuring that the extracted compounds retain their functionality and efficacy over time is crucial for the commercial success of the products.

Economic Viability: Assessing the overall economic viability of large-scale UAE involves balancing extraction yields, energy costs, equipment investments, and market demand. A comprehensive cost–benefit analysis is essential to determine the feasibility and competitiveness of the industrial-scale process.

Waste Management: Dealing with larger volumes of waste generated during the extraction process requires efficient waste management systems. Addressing environmental concerns and complying with waste disposal regulations is vital for sustainable industrial practices.

In conclusion, while ultrasound-assisted extraction shows great promise in small-scale studies, successfully transitioning to industrial applications requires overcoming various challenges related to cost, scalability, process optimization, and regulatory compliance.

*3.7. Cost-Effectiveness and Feasibility of Using Rosa canina Extracts for Sustainable Dyeing and Functional Finishing of Cotton Fabrics*

Concerning the cost-effectiveness and feasibility of using *Rosa canina* extracts for sustainable dyeing and functional finishing of cotton fabrics, there are some potential aspects:

Raw Material Accessibility: *Rosa canina* is a wild plant found in various regions. If the raw material is locally accessible, it can contribute to cost-effectiveness by reducing transportation costs.

Low-Cost Extraction Methods: Traditional extraction methods, such as water-based methods, are generally cost-effective compared to more complex or energy-intensive techniques. Using these methods can contribute to the overall economic viability.

Bath Exhaustion Method: The application of *Rosa canina* extracts via the bath exhaustion method is a common and straightforward technique. It can be cost-effective in terms of application simplicity and reduced energy consumption.

Mordanting Process: The use of chitosan as a bio-mordant may contribute to cost-effectiveness, as chitosan is derived from chitin, a natural and abundant polymer found in crustacean shells.

Color Fastness and Stability: The challenge lies in achieving consistent colorfastness and stability on cotton fabrics, which may require additional processes or treatments.

Quality Control: Maintaining consistent quality in *Rosa canina* extracts, which can vary based on factors like plant maturity and extraction conditions, may require rigorous quality control measures.

Technical Feasibility: Utilizing *Rosa canina* extracts for dyeing and finishing is technically feasible, especially given the historical use of plant-based dyes. However, consistent results at an industrial scale require optimization.

Market Demand: Feasibility is linked to market demand. As there is a growing market preference for sustainable and plant-based textiles, the use of *Rosa canina* extracts becomes more feasible.

Environmental Impact: If *Rosa canina* extracts replace synthetic dyes, especially those derived from petrochemicals, the overall environmental impact could be reduced, enhancing the feasibility from a sustainability perspective.

In conclusion, the cost-effectiveness and feasibility of using *Rosa canina* extract for sustainable dyeing and functional finishing of cotton fabrics depend on optimizing the extraction process, addressing technical challenges, and aligning with market demands for eco-friendly textiles. Additionally, ensuring regulatory compliance and considering the environmental impact contributes to the overall feasibility of this approach.

## 4. Conclusions

Among the fabric samples tested, *Rosa canina* dry fruit water extraction-treated cotton fabrics (Sample 4) demonstrated the highest UV protection, achieving a UPF 50 value, classifying it as having excellent UV protection. Notably, chitosan+ *Rosa canina* dry fruit water extraction-treated fabrics (Sample 1) exhibited both UV protection and antibacterial activity against Staphylococcus aureus, attributed to the presence of chitosan and *Rosa canina* extraction. Overall, using water as a solvent in *Rosa canina* extractions proved more effective in imparting desirable functionality properties on cotton fabrics compared to methanol or water/methanol mixtures. The use of dry fruits for extraction provided advantages over fresh fruits, supported by UV-Vis and HPLC-UV analyses revealing a greater variety of active compounds in dry extractions. The color measurement results indicated a slight color change, particularly in yellowness (b), with methanol-treated fabrics. The highest K/S values at 400 nm were achieved with fabrics treated with *Rosa canina* extractions using water. Pre-treatment with chitosan did not significantly impact the color change or color strength of cotton fabrics, although all treated samples exhibited light colors on the L* axis.

Future studies will focus on refining the *Rosa canina* extraction process for increased effectiveness and exploring its application to wool and synthetic fabrics, particularly Polyamide (PA). Subsequent investigations will include color measurements, fastness properties, antimicrobial and UV protection properties, washing durability, and mechanical properties of treated fabrics, as well as the overall durability of *Rosa canina* extraction treatments.

In conclusion, *Rosa canina* extracts present a compelling option for sustainable dyeing and functional finishing of textiles, making notable contributions to a greener textile industry. The abundance of *Rosa canina* in regions where it naturally grows, coupled with the potential cost-effectiveness of ultrasound-assisted extraction, positions it as an economically viable choice. Furthermore, apart from the UV protection and antibacterial properties, the extracts exhibit potential medicinal and antioxidant properties, adding value beyond mere coloring. In terms of environmental impact, *Rosa canina* extracts outshine some synthetic dyes, aligning with the growing demand for eco-friendly practices. However, challenges exist, such as the need for optimization in colorfastness and stability, ensuring consistency in extraction quality, and addressing the processing time and energy consumption associated with ultrasound-assisted extraction. Overcoming these challenges

could establish *Rosa canina* extracts as a sustainable and beneficial alternative for the textile industry, blending ecological consciousness with functional versatility and representing a significant stride toward a greener textile industry.

**Author Contributions:** Conceptualization, R.A. and M.B.-A.; methodology, R.A. and M.B.-A.; software, R.A. and I.M.-G.; validation, R.A. and M.B.-A.; formal analysis, R.A.; investigation, R.A. and M.B.-A.; resources, R.A. and M.B.-A.; data curation, R.A. and I.M.-G.; writing—original draft preparation, R.A.; writing—review and editing, R.A. and I.M.-G.; visualization, R.A. and I.M.-G.; supervision, M.B.-A. and P.D.-G.; project administration, R.A., M.B.-A. and P.D.-G.; funding acquisition, R.A. and P.D.-G. All authors have read and agreed to the published version of the manuscript.

**Funding:** This study was funded by The Scientific and Technological Research Council of Turkey (TUBITAK) 2219-International Postdoctoral Research Fellowship Program for Turkish Citizens.

**Institutional Review Board Statement:** Not applicable.

**Informed Consent Statement:** Not applicable.

**Data Availability Statement:** The data related to the results of this study are available upon request from the corresponding author.

**Acknowledgments:** All experiments were conducted in the laboratories of Universidad Politécnica de Valencia—Department of Textile and Paper Engineering. We also would like to send our special thanks to "Iván Sciscenko" for performing HPLC-UV analysis.

**Conflicts of Interest:** The authors declare no conflicts of interest. The funders had no role in the design of the study; in the collection, analyses, or interpretation of data; in the writing of the manuscript; or in the decision to publish the results.

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
