# Peer review of "Sustainable Dyeing and Functional Finishing of Cotton Fabric by Rosa canina Extracts"

_sustainability, doi:10.3390/su16010227_

Round 1
Reviewer 1 Report
Comments and Suggestions for Authors
Please find attached report

This work deserves publication after corrections
Author Response
AUTHORS’ REPLY
First of all, we would like to thank the reviewers for their constructive criticism, informative comments and valuable suggestions about our manuscript entitled “Sustainable Dyeing and Functional Finishing of Cotton Fabric by Rosa Canina Extracts”. We carefully considered every point of the reviewers, and made the necessary changes/corrections in the manuscript.
- Please provide the detailed information in Data Availability Statement (DAS) at the back section.
“Data Availability Statement” has been added at the back section.
2. Please revise the repeated sentences according to the attached file.
Entire manuscript has been revised according to English language grammar, and the similarity.
- We noticed that this paper lacks of table 4 caption, please add.
It has been added.
REVİEWER1:
It is well known to us, and since I have been a specialist in the sciences of dyeing, printing, and fabric processing for thirty-five years, that cotton fabric does not respond to the dyeing process, especially with extracted natural dyes, except after the mordating process, which the authors carried out using chitosan not only to complete the dyeing process only, but I am trying to give the fabric some properties, including giving it an active property against bacteria.
Although the idea was developed by many researchers, the method of presentation and the amount of data presented make this work suitable for publication, but only after making important modifications, including, for example:
1- The title must replace the word “coloring” with the word “dyeing.”
The title has been changed as “ Sustainable Dyeing and Functional Finishing of Cotton Fabric by Rosa Canina Extracts”.
2- On line 214, write in detail, not briefly, about UV Protection of Fabrics.
The “2.8 UV Protection of Fabrics” section was rewritten in detail.
3- On line 222, write in detail, not briefly, about Antibacterial Activity of Fabrics, then the word (ATCC 6538) on line 225 must be replaced with the word (AATCC 6538).
It has been replaced.
4- In line 236, it is preferable to write the Kubelkae-Munk equation as follows.
It has been changed.
5- On line 431, Table 3 (L, a, b), you must write as follows: L*, a*, b*
It has been corrected.
6- The English language and grammar must be reviewed more carefully.
Entire manuscript has been revised according to English language and grammar.
7- I offer advice to the authors and not to criticize the study. Why did you perform the process of extracting the natural color when using 60 minutes? In our published research, we conduct a study of time, meaning that we work at different times. Who knows, when using a time of 90 or 120 minutes, you will obtain a natural color extracted with a stronger color. Therefore, the authors only have to discuss the time factor even though they did not focus on using only 60 minutes in the extraction process.
Thank you for your valuable suggestion. We have tried different times as 30 min, 45 min and 60 min in our previous studies (see ref. 57 and 58).
- R. Atakan; J. Gisbert-Payá; P. Díaz-García and M. Bonet-Aracil. Obtención de fitocomponentes de Rosa Cania a partir de la extracción con ultrasonidos. In Proceeedings of the IX. Congreso I+D+i Campus de Alcoi. Creando Sinergias, Alcoy, Spain, 2022, pp. 161–164.
- R. Atakan; P. Díaz-García; J. Gisbert-Payá and M. Bonet-Aracil. Use of Rosa Canina Extractions for Ecofriendly Textile Finishing. In Proceedings of AUTEX2023- The World Textile Conference, Melbourne, Australia, 2023.
60 minutes gave the best results among others. Yes, we have never tried longer minutes for extraction and that would be a great idea. However, in our further studies, we increased the temperature as 80ºC and 100ºC, also found excellent UPF results. As we continue our experiments, we will be sharing all further results in different academic platforms.
8- References must be added to recent, relevant references. Please put these two references in the references:
- M. El-Apasery, A. Hussein, Mohamed Saleh, Abubaker A Eladasy, Microwave-assisted dyeing of wool fabrics with natural dyes as eco- friendly dyeing method: part I. Dyeing performance and fastness properties, 2021, Egyptian Journal of Chemistry 64 (7), 3751-3759.
- MA El-Apasery, AM Hussein, NM Nour El-Din, MO Saleh, ABA El-Adasy, Microwave-assisted dyeing of wool fabrics with natural dyes as eco-friendly dyeing method: part II. The effect of using different mordants Egyptian Journal of Chemistry 2021, 64 (7), 3761-3766.
They have been added as ref. 53 and 54.
Reviewer 2 Report
Comments and Suggestions for Authors
1-you are used cotton fabric in your work but you are not mentioned the source of it and the name of company where you bring it, please write it in material section.
2-at page 4 line 152 what is meaning of'"solvent concentration" 200 g/L?
3-at page 4 line 188 sentence not start with letter A.
4-at page 5 line 205 you are treated cotton fabrics with 1 g/L of: vanillic, caffeic, ferulic, gallic, p-couramic and protocautehauic acids but you neglect them in your discussion why?
5-at figure 3 you write yellow, blue and green please change it to symbol because the manuscript are colourless.
6-at figure 6 the peaks not appear please re-draw again.
7-you are made three extract with water,methanol and water/methanol but you are neglect the treated chitosan samples with water/methanol in your discussion why?
8- color measurement section need to rewrite again in good form (discussion at table 3 and figure 8).
9-at antibacterial results why the samples treated with ch.+R.C.dry water ext. and ch.+R.C dry methanol ext. give the high antibacterial results please I need scientific explanation.
Comments on the Quality of English Language
check the grammar rules
Author Response
AUTHORS’ REPLY
First of all, we would like to thank the reviewers for their constructive criticism, informative comments and valuable suggestions about our manuscript entitled “Sustainable Dyeing and Functional Finishing of Cotton Fabric by Rosa Canina Extracts”. We carefully considered every point of the reviewers, and made the necessary changes/corrections in the manuscript.
- Please provide the detailed information in Data Availability Statement (DAS) at the back section.
“Data Availability Statement” has been added at the back section.
2. Please revise the repeated sentences according to the attached file.
Entire manuscript has been revised according to English language grammar, and the similarity.
- We noticed that this paper lacks of table 4 caption, please add.
It has been added.
REVİEWER2:
1-you are used cotton fabric in your work but you are not mentioned the source of it and the name of company where you bring it, please write it in material section.
The necessary information has been added into “2.1. Material” section.
2- at page 4 line 152 what is meaning of '"solvent concentration" 200 g/L?
It has been changed as fruit concentration. 200g/L means: 1 L solvent contains 200 g fruits.
3- at page 4-line 188 sentence not start with letter A.
It has been corrected.
4- at page 5 line 205 you are treated cotton fabrics with 1 g/L of: vanillic, caffeic, ferulic, gallic, p-couramic and protocautehauic acids but you neglect them in your discussion why?
Thank you for your warning. Yes, we treated cotton fabrics with 1 g/L of: vanillic, caffeic, ferulic, gallic, p-couramic and protocautehauic acids and tested them in terms of UPF values to identify if any of them responsible of UV protection property. Caffeic, ferulic and p-couramic acids were found to provide UV protection with UPF values of 1000+, 357 and 301, respectively.
However, in our extractions, we could not detect any of the standard solutions (vanillic, caffeic, ferulic, gallic, p-couramic and protocautehauic acids). So, we neglected to discuss this detail in the manuscript. We also removed these extra experiments in “2.6. Textile Dyeing & Finishing Process” section.
5- at figure 3 you write yellow, blue and green please change it to symbol because the manuscript are colourless.
They have been removed.
6- at figure 6 the peaks not appear please re-draw again.
It has been changed.
7- you are made three extracts with water, methanol and water/methanol but you are neglect the treated chitosan samples with water/methanol in your discussion why?
We have not studied treated chitosan samples with water/methanol extractions. As we seen UV Protection and antibacterial results represented in Table 4, water as a solvent provides the most desirable results compared to methanol solvent. Results provided with water/methanol solvents are almost the average of these two. (See the results of these samples: 4,5,6,7,8 and 9).
UPF of Chitosan+ R.C. (dry) water ext. treated: 37.57
UPF of Chitosan+ R.C. (dry) methanol ext. treated: 16,36
UPF of Chitosan+ R.C. (fresh) water ext. treated: 34,64
UPF of Chitosan+ R.C. (fresh) methanol ext. treated: 15,78
Therefore, UPF of chitosan + R.C water/methanol extractions would be ~26.
8- color measurement section needs to rewrite again in good form (discussion at table 3 and figure 8).
It has been revised and rewritten.
9- at antibacterial results why the samples treated with Ch.+R.C.dry water ext. and Ch.+R.C dry methanol ext. give the high antibacterial results please I need scientific explanation.
Discussion of Antibacterial results section has been revised and rewritten.
10- Comments on the Quality of English Language: check the grammar rules
Entire manuscript has been revised according to English language and grammar.
Reviewer 3 Report
Comments and Suggestions for Authors
This paper studied the effects of processing parameters on the extraction of Rosa Canina. Obtained extractions were further investigated by various analytical methodology. Have the feeling that there is lack of in-depth discussions in terms of the extracted compounds. Therefore, major revisions are needed before publication:
1. Line 109: Please define UPF upon its first occurrence.
2. Line 243 to 244: “The UV-Vis absorption spectroscopy of Rosa Canina extractions with different solvents provides an amount of detail about their active compounds….” Can the author provide the details of these active compounds, what there are according to UV spectroscopy, at least provide some speculations?
3. Line 245: what is “waste solutions”? can the authors provide more details about how is “waste solutions” created during extraction? Is its amount quantifiable?
4. Line 257” “which confirms that this bond can only be occurred in the presence of alcohol and water” what is this bond? Why can only be occurred in alcohol and water? Any speculations?
5. Line 274 to 279: since the absorption peak is higher in dry fruit than that in fresh fruit, then why “color features of methanol extractions are stronger in fresh fruit extractions than dry fruit ones”? Can the authors provide more details?
6. Line 298 to 299: why lower intensity lead to slightly stronger bonds? Any detailed explanations?
7. Line 319 to 320: “still has at least one active compound of its original extraction with a stronger bond”? can the authors elaborate on this?
8. For the HPLC-UV data, can the authors provide speculations about the possible chemistry of extracted components according to the peaks?
Author Response
AUTHORS’ REPLY
First of all, we would like to thank the reviewers for their constructive criticism, informative comments and valuable suggestions about our manuscript entitled “Sustainable Dyeing and Functional Finishing of Cotton Fabric by Rosa Canina Extracts”. We carefully considered every point of the reviewers, and made the necessary changes/corrections in the manuscript.
- Please provide the detailed information in Data Availability Statement (DAS) at the back section.
“Data Availability Statement” has been added at the back section.
2. Please revise the repeated sentences according to the attached file.
Entire manuscript has been revised according to English language grammar, and the similarity.
- We noticed that this paper lacks of table 4 caption, please add.
It has been added.
REVİEWER3:
This paper studied the effects of processing parameters on the extraction of Rosa Canina. Obtained extractions were further investigated by various analytical methodology. Have the feeling that there is lack of in-depth discussions in terms of the extracted compounds. Therefore, major revisions are needed before publication:
1.Line 109: Please define UPF upon its first occurrence.
It has been added.
2.Line 243 to 244: “The UV-Vis absorption spectroscopy of Rosa Canina extractions with different solvents provides an amount of detail about their active compounds….” Can the author provide the details of these active compounds, what there are according to UV spectroscopy, at least provide some speculations?
Yes sure. We have added some detail info about it into “2.3. Analysis of Extract Solutions by UV-Vis Spectrophotometry” section.
By UV-Vis spectroscopy, we could simply get information about the amount and variety of active compounds that can be possibly transferred to the fabrics. Unfortunately, UV-Vis analysis is not sufficient analytical method to identify the exact chemical component and the exact amount of this chemical component. We performed this analysis just to predict the best one and be able to make a comparison between extractions, as we conducted several extraction processes.
According to other scientific studies below (ref. 55 and 56) regarding to identify active compounds present in Rosa Canina extractions, we expect to detect some phenolic compounds such as “Citric acid, Gallacetophenone, Procyanidin, 2,3-Digalloylglucose, Pyrogallol-2-O-glucuronide, Catechin, Gallocatechol, Quercetin 3,7-diglucoside, Quercetin dihydrate, Bilobalide, Luteolin 5-methyl ether, Quercetol 3-O-rutinoside, Ellagic acid, Kaempferol-7-O-glucoside, Quercetin, Rosmarinic acid, Apigenin, Resveratrol etc.”. However, to identify the chemical constituents of extractions, the most efficient techniques are HPLC-DAD/-ESI-MS and HPLC-UV-MS. HPLC or UPLC with a combination of MS seems to be ideal for identification of phenols in Rosa Canina extractions.
- A. Stănilă et al., “Extraction and characterization of phenolic compounds from rose hip (Rosa canina L.) using liquid chromatography coupled with electrospray ionization - mass spectrometry,” Not. Bot. Horti Agrobot. Cluj-Napoca, vol. 43, no. 2, pp. 349–354, 2015.
- S. Fetni, N. Bertella, A. Ouahab, J. M. Martinez Zapater, and S. De Pascual-Teresa Fernandez, “Composition and biological activity of the Algerian plant Rosa canina L. by HPLC-UV-MS,” Arab. J. Chem., vol. 13, no. 1, pp. 1105–1119, 2020.
We have given all details about the extractions processes and potential active compounds that could be obtained in our review paper entitled “Rosa Canina Extractions as a Natural Dye and Finishing Agent for Ecofriendly Textile Applications - A review”, which is under evaluation process in another journal right now.
3.Line 245: what is “waste solutions”? can the authors provide more details about how is “waste solutions” created during extraction? Is its amount quantifiable?
Yes sure. Related information has been added into “2.6. Textile Dyeing & Finishing Process” section. Yes, the amount was quantifiable, we kept ~100 mL of waste dye solutions for UV-Vis analysis.
4.Line 257” “which confirms that this bond can only be occurred in the presence of alcohol and water” what is this bond? Why can only be occurred in alcohol and water? Any speculations?
That is only a strong prediction. The only differences of extractions are solvents: water, methanol (a type of alcohol) and water + methanol (alcohol). If there is an extra peak in UV-Vis graphs of water + methanol (alcohol), it is most probably occurred in the presence of water and alcohol.
Because, the appearance of an extra peak suggests the potential presence of a compound or compounds that are soluble in the combined water and methanol solvent but not in water or methanol alone. The choice of solvent in the extraction process plays a critical role in the solubility of different compounds. The water + methanol mixture may have enhanced solubilizing properties compared to the individual solvents, leading to the extraction of additional compounds. This could include polar compounds that are not effectively extracted by water or nonpolar compounds not efficiently extracted by methanol alone. However, identifying the compound responsible for the extra peak would require further analysis, potentially using additional techniques such as chromatography.
- Line 274 to 279: since the absorption peak is higher in dry fruit than that in fresh fruit, then why “color features of methanol extractions are stronger in fresh fruit extractions than dry fruit ones”?
Thanks for warning and good point. Yes, we have changed the sentence as “color features of methanol extractions are stronger in dry fruit extractions than fresh fruit ones”. This has been confirmed by the DE*ab values (color change) of treated samples (seen in Table 3), as well:
DE*ab values of R.C. fresh fruit methanol ext. treated: 9,2075
DE*ab values of R.C. dry fruit methanol ext. treated: 12,446 (is higher than fresh fruit one).
- Line 298 to 299: why lower intensity lead to slightly stronger bonds? Any detailed explanations?
The explanation has been changed as: “In waste solutions the bonds are slightly stronger than original extraction due to presence of unreacted extract components, possible impurities from cotton fabric, residual of chitosan.”
Particularly, waste dye solutions may also contain:
-Unreacted Extract Components: Some components of the natural extract may not fully bind to the fabric and may be present in the waste solution.
-Impurities from Cotton Fabric: Substances released from the cotton fabric itself during the dyeing process.
-Residue of Mordant: Mordant (Chitosan) used in the dyeing process to enhance color fixation or modify dye properties.
Therefore, these unreacted extract components, impurities from cotton fabric and residue of chitosan may lead new bonds or higher peaks.
- Line 319 to 320: “still has at least one active compound of its original extraction with a stronger bond”? can the authors elaborate on this?
The explanation has been changed as: “In Figure 5b, only one sharp adsorption peak with an intensity of 4.5 was observed for extraction at 320 nm, however waste solution shows two different bonds with intensities 3.7 and 4.1 at 340 nm and 315 nm, respectively. That result indicate that waste solution of fresh Rosa Canina methanol/water extraction still has some unreacted active compounds, methanol, water and also impurities of cotton fabric, which can lead new bonds with each other”.
- For the HPLC-UV data, can the authors provide speculations about the possible chemistry of extracted components according to the peaks?
HPLC-UV data cannot provide sufficient information to identify or predict about the possible chemistry components. It can only show: if the same chemicals, we introduced to the system, present in the extraction, too. According to our HPLC-UV data, our extractions do not consist of protocatechuic acid vanillic acid, caffeic acid, p-couramic acid or ferulic acid, which we introduced them to the device as standard solutions.
In addition, The High-Performance Liquid Chromatography with Ultraviolet Detection (HPLC-UV) chromatograms can vary based on the type of column used in the chromatographic system. We used C18 column, so different column may lead different retention time and peak intensities. Because, different columns, even within the C18 category, can have variations in their stationary phase properties, such as particle size, pore size, and bonding chemistry. These differences can influence the retention time and peak intensities of analytes during chromatographic separation. Therefore, when changing the column, it is expected to observe differences in the chromatographic profile, including changes in retention times and peak shapes.
HPLC-UV needs to be coupled with other detectors, such as mass spectrometry (HPLC-MS), to provide complementary information and enhance compound identification capabilities.
Reviewer 4 Report
Comments and Suggestions for Authors
The article explores the sustainable coloration and functional finishing of cotton fabrics using Rosa Canina extracts through a novel ultrasound-assisted extraction process. Both dry and fresh fruits are considered, employing various solvents such as distilled water, methanol, and water/methanol (50/50% v/v) in the extraction process, conducted at 60°C for 60 minutes. The extracted compounds are analyzed using Ultraviolet–visible (UV-Vis) spectroscopy and High-performance liquid chromatography incorporated with ultra-violet spectroscopy (HPLC-UV) to gain a comprehensive understanding of their chemical composition. The study aims to identify active compounds and explore their potential impact on the functionality of treated fabrics. Following extraction, cotton fabrics undergo a one-step application process using a biomordant, chitosan, in a pre-mordanting process. Fourier transform infrared spectroscopy (FTIR) is utilized to analyze the surface chemistry and chemical composition of treated fabrics. The article evaluates treated and untreated fabrics, both with and without mordant, in terms of UV protection and antibacterial properties. Color measurements are conducted to assess fabric color changes and dyeability properties. Additionally, waste solutions from textile applications are analyzed by UV-Vis spectroscopy to observe potential transfer of active compounds to fabrics. The results indicate that Rosa Canina extracts exhibit significant potential for sustainable coloration and functional finishing of cotton fabrics. Overall, the article demonstrates significant potential for publication in the Sustainability journal with the suggested major revision.
· Abstract, consider addressing the extraction process, chemical analysis, textile applications, and results. All this has to be elaborated in the abstract.
· Abstract could be more specific about the identified active compounds and their potential impact on fabric functionality. Enhancing clarity in these aspects will improve the abstract's overall readability and impact.
· Generally, there is a lack of information on the specific active compounds present in Rosa Canina extracts and their concentrations.
· Limited discussion on the extraction process parameters such as ultrasound intensity, extraction time, and solvent ratios, which could affect the efficiency of extraction. The authors are recommended to improve.
· Further explanation of the bio mordant chitosan and its role in the textile application process.
· The authors are suggested to discuss the specific UV protection properties and antibacterial properties tested on the treated cotton fabrics.
· More analysis of the color changes on fabrics and the dyeability properties of the extractions might improve the presented work.
· Elaborate on the specific active compounds present in the waste solutions of textile applications and their potential environmental impact in the introduction part. Make sure not make the introduction shorter.
· Absence of comparison with other natural dye sources or synthetic dyes to evaluate the superiority of Rosa Canina extracts.
· Insufficient discussion on the potential limitations or challenges in scaling up the ultrasound-assisted extraction process for industrial applications.
· Elaborate on the potential cost-effectiveness and feasibility of using Rosa Canina extracts for sustainable coloration and functional finishing of cotton fabrics.
· Limited discussion on the overall sustainability aspects of the proposed approach and its contribution to a greener textile industry.
Addressing these comments will improve the presented work.
Comments on the Quality of English Language
Minor English correction required.
Author Response
AUTHORS’ REPLY
First of all, we would like to thank the reviewers for their constructive criticism, informative comments and valuable suggestions about our manuscript entitled “Sustainable Dyeing and Functional Finishing of Cotton Fabric by Rosa Canina Extracts”. We carefully considered every point of the reviewers, and made the necessary changes/corrections in the manuscript.
- Please provide the detailed information in Data Availability Statement (DAS) at the back section.
“Data Availability Statement” has been added at the back section.
2. Please revise the repeated sentences according to the attached file.
Entire manuscript has been revised according to English language grammar, and the similarity.
- We noticed that this paper lacks of table 4 caption, please add.
It has been added.
REVİEWER4:
The article explores the sustainable coloration and functional finishing of cotton fabrics using Rosa Canina extracts through a novel ultrasound-assisted extraction process. Both dry and fresh fruits are considered, employing various solvents such as distilled water, methanol, and water/methanol (50/50% v/v) in the extraction process, conducted at 60°C for 60 minutes. The extracted compounds are analyzed using Ultraviolet–visible (UV-Vis) spectroscopy and High-performance liquid chromatography incorporated with ultra-violet spectroscopy (HPLC-UV) to gain a comprehensive understanding of their chemical composition. The study aims to identify active compounds and explore their potential impact on the functionality of treated fabrics. Following extraction, cotton fabrics undergo a one-step application process using a biomordant, chitosan, in a pre-mordanting process. Fourier transform infrared spectroscopy (FTIR) is utilized to analyze the surface chemistry and chemical composition of treated fabrics. The article evaluates treated and untreated fabrics, both with and without mordant, in terms of UV protection and antibacterial properties. Color measurements are conducted to assess fabric color changes and dyeability properties. Additionally, waste solutions from textile applications are analyzed by UV-Vis spectroscopy to observe potential transfer of active compounds to fabrics. The results indicate that Rosa Canina extracts exhibit significant potential for sustainable coloration and functional finishing of cotton fabrics. Overall, the article demonstrates significant potential for publication in the Sustainability journal with the suggested major revision.
1-Abstract, consider addressing the extraction process, chemical analysis, textile applications, and results. All this has to be elaborated in the abstract.
Sure, Abstract has been revised and rewritten accordingly.
2-Abstract could be more specific about the identified active compounds and their potential impact on fabric functionality. Enhancing clarity in these aspects will improve the abstract's overall readability and impact.
Sure, Abstract has been revised and rewritten accordingly.
3-Generally, there is a lack of information on the specific active compounds present in Rosa Canina extracts and their concentrations.
Yes, UV-Vis and HPLC-UV has limited capacity to identify active compounds. HPLC-UV data cannot provide sufficient information to identify or predict about the possible chemistry components. It can only show: if the same chemicals, we introduced to the system, are present in the extraction, too. According to our HPLC-UV data, our extractions do not consist of protocatechuic acid vanillic acid, caffeic acid, p-couramic acid or ferulic acid, which we introduced them to the device as standard solutions.
According to other scientific studies below (ref. 55 and 56) regarding to identify active compounds present in Rosa Canina extractions, we expect to detect some phenolic compounds such as “Citric acid, Gallacetophenone, Procyanidin, 2,3-Digalloylglucose, Pyrogallol-2-O-glucuronide, Catechin, Gallocatechol, Quercetin 3,7-diglucoside, Quercetin dihydrate, Bilobalide, Luteolin 5-methyl ether, Quercetol 3-O-rutinoside, Ellagic acid, Kaempferol-7-O-glucoside, Quercetin, Rosmarinic acid, Apigenin, Resveratrol etc.”
- A. Stănilă et al., “Extraction and characterization of phenolic compounds from rose hip (Rosa canina L.) using liquid chromatography coupled with electrospray ionization - mass spectrometry,” Not. Bot. Horti Agrobot. Cluj-Napoca, vol. 43, no. 2, pp. 349–354, 2015.
- S. Fetni, N. Bertella, A. Ouahab, J. M. Martinez Zapater, and S. De Pascual-Teresa Fernandez, “Composition and biological activity of the Algerian plant Rosa canina L. by HPLC-UV-MS,” Arab. J. Chem., vol. 13, no. 1, pp. 1105–1119, 2020.
HPLC-UV needs to be coupled with other detectors, such as mass spectrometry (HPLC-MS), to provide complementary information and enhance compound identification capabilities. We have no fundings to perform UPLC-MS or HPLC-MS spectrometry at this moment.
4-Limited discussion on the extraction process parameters such as ultrasound intensity, extraction time, and solvent ratios, which could affect the efficiency of extraction. The authors are recommended to improve.
Thank you for your valuable suggestion. We have performed ultrasound-assisted extractions at different temperatures (30ºC, 45ºC and 60ºC) for different times (30 min, 45 min and 60 min) with different fruit concentrations (50 g/L and 200g/L) in our previous studies (see ref. 57 and 58). As we have a basic ultrasound device (P-Selecta Ultrasons H-D device) it has the intensity of 40 kHz.
- R. Atakan; J. Gisbert-Payá; P. Díaz-García and M. Bonet-Aracil. Obtención de fitocomponentes de Rosa Cania a partir de la extracción con ultrasonidos. In Proceeedings of the IX. Congreso I+D+i Campus de Alcoi. Creando Sinergias, Alcoy, Spain, 2022, pp. 161–164.
- R. Atakan; P. Díaz-García; J. Gisbert-Payá and M. Bonet-Aracil. Use of Rosa Canina Extractions for Ecofriendly Textile Finishing. In Proceedings of AUTEX2023- The World Textile Conference, Melbourne, Australia, 2023.
60ºC and 60 minutes gave the best results among others. Yes, we have never tried longer minutes for extraction and that would be a great idea. As we continue our studies, we have increased the temperature as 80ºC and 100ºC, (keeping the other parameters stable: 60 min and 200 g/L fruit conc.) also found excellent UPF results for both cotton and PA fabrics. We will be sharing all further results in different academic platforms or papers.
5-Further explanation of the bio mordant chitosan and its role in the textile application process.
More explanations of bio mordant chitosan in terms of its role in the textile applications, and its importance for sustainability as well as related reference papers have been added in the introduction and discussion of UV protection and antibacterial activity parts.
6-The authors are suggested to discuss the specific UV protection properties and antibacterial properties tested on the treated cotton fabrics.
The discussion section of UV Protection and Antibacterial Results has been revised and improved.
7-More analysis of the color changes on fabrics and the dyeability properties of the extractions might improve the presented work.
Many thanks for your valuable contribution. The color analysis section has been revised and improved.
8-Elaborate on the specific active compounds present in the waste solutions of textile applications and their potential environmental impact in the introduction part. Make sure not make the introduction shorter.
Thank you for your valuable suggestion. Related information has been added in the introduction part.
9-Absence of comparison with other natural dye sources or synthetic dyes to evaluate the superiority of Rosa Canina extracts.
Adding all these comparisons makes this research article too comprehensive. We explained all these aspects in our review paper entitled “Rosa Canina Extractions as a Natural Dye and Finishing Agent for Ecofriendly Textile Applications - A review”, which is under evaluation process in another journal right now.
However, we have added some critical points in the conclusion section.
10-Insufficient discussion on the potential limitations or challenges in scaling up the ultrasound-assisted extraction process for industrial applications.
Another section has been added in to Discussion Part: “3.6. Scaling Up Ultrasound-Assisted Rosa Canina Extraction for Industrial Applications”
11-Elaborate on the potential cost-effectiveness and feasibility of using Rosa Canina extracts for sustainable coloration and functional finishing of cotton fabrics.
Another section has been added in to Discussion Part: “3.7. Cost-effectiveness and Feasibility of using Rosa Canina Extracts for Sustainable Dyeing and Functional Finishing of Cotton Fabrics”.
12-Limited discussion on the overall sustainability aspects of the proposed approach and its contribution to a greener textile industry.
Overall sustainability aspects of the proposed approach have been added in the conclusion section.
Round 2
Reviewer 2 Report
Comments and Suggestions for Authors
thank you you done my comments
Reviewer 3 Report
Comments and Suggestions for Authors
Recommend for publication
Reviewer 4 Report
Comments and Suggestions for Authors
Authors responded to all comments. The manuscript got revised and improved according to the comments and hence, the current version of the manuscript can be accepted in Sustainability Journal
Comments on the Quality of English LanguageMinor English corrections needed.